# CMIRA: Class Membership Inducing Recovery Attacks Against Machine Unlearning Models

## Abstract

The implementation of data privacy regulations such as GDPR and CCPA has advanced machine learning (MU) technology, which is designed to facilitate the removal of specific sensitive data points from trained models upon request. Despite rapid advancements in MU technology, its vulnerabilities are still underexplored, posing potential risks of privacy breaches by recovering unlearned sensitive information. Further, existing research on MU vulnerabilities often requires access to the original models, which violates the core objective of MU. To address this gap, we reformulate the study of attacks against released unlearned models and present the first work to explore recovery attacks on MU models without requiring access to the original model. Our approach, known as Class Membership Inducing Recovery Attack (CMIRA), effectively recovers forgotten data by exploiting a probing dataset. Specifically, we implement the CMIRA scheme regarding mutual knowledge distillation between MU and attack models. Extensive experiments across multiple datasets and MU methods demonstrate that CMIRA exhibits high efficacy in both theoretical analysis and practical applications. Our study highlights the need for developing more robust MU systems and lays the groundwork for future research to establish new benchmarks for evaluating their security.

## 1 Introduction

The emergence of machine unlearning (MU) is driven by stringent data privacy regulations such as GDPR (Hoofnagle et al., 2019) and CCPA (Itakura & Terada, 2018), which require the removal of specific sensitive data upon request. MU is designed to forget particular data points from the learned models (Cao & Yang, 2015). As concerns about the increasing data misuse and privacy breaches, MU has gained more attention as a critical component in building safe machine learning systems.

Despite rapid advancements in MU techniques (Guo et al., 2023; Bourtoule et al., 2021; Wu et al., 2020; Brophy & Lowd, 2020; Gupta et al., 2021; Chen et al., 2021a; Thudi et al., 2021; 2022), the study of their vulnerabilities (Hu et al., 2024) remains underexplored. This oversight poses a potential risk of privacy breaches by recovering information about forgotten data, highlighting the limited research to date on the full scope of MU vulnerabilities. The only existing research (Hu et al., 2024) that investigates attacks against MU models was recently published. However, this work is based on the impractical assumption that unlearning inversion attacks require access to both originally learned and unlearned models, as illustrated in Figure 1. In general, MU aims to release an unlearned model in which the correct map $\mathcal{D}_f : \mathcal{X}_f \mapsto \mathcal{Y}_f$ has been distorted from the original model. It is imperative that the originally learned model is inaccessible to users, as such a violation may significantly increase the risk of privacy breaches.

To advance research in this area, we reformulate the study of attacks against released unlearned models, eliminating the need to access the original models. As demonstrated in Figure 1, our objective is to design an attack model to recover the correct output $\mathcal{Y}_f$ given the input data $\mathcal{X}_f$ and the unlearned model. As most MU models (Bourtoule et al., 2021) restrict the scope of the investigation to the mature unlearning area of image classification tasks, our study is conducted in a similar way on these MU models. Inspired by the membership inference attack (MIA) (Shokri et al., 2017) against machine learning models, we propose a class membership inducing recovery attack (**CMIRA**) scheme against machine unlearning models.

Figure 1: The demonstration of recovery attack against MU models, which is critical and prospective to study the vulnerability of current MU-based privacy preservation.

Analogous to the shadow training sets (Shokri et al., 2017) for MIA, we create a probing dataset $\mathcal{D}_p : \mathcal{X}_p \mapsto \mathcal{Y}_p$ that is similar to $\mathcal{D}_f$ for CMIRA. Note that $\mathcal{D}_p$ can be easily created by finding similar images and their labels through image search engines given the query images $\mathcal{X}_f$. In particular, we design a mutual knowledge distillation (MKD) approach in which the attack model $\mathcal{M}_A$ can iteratively recover plausible knowledge, that is, $\mathcal{X}_f \mapsto \hat{\mathcal{Y}}_f$, by inducing the unlearned model $\mathcal{M}_U$ to recover the class memberships of $\mathcal{X}_f$ with the knowledge distilled from $\mathcal{D}_p$.

We summarize our contributions as follows: ❶ To the best of our knowledge, this is the *first attempt* to study the recovery attacks against increasingly used MU models, which can effectively assess the risk of data privacy breaches and promote the robustness of MU study. ❷ We propose CMIRA, a MU model-agnostic attack scheme, to effectively recover the true class memberships from mostly used MU models. ❸ We implement CMIRA with a recovery attack model to recover the class memberships from MU models via mutual knowledge distillation based on a probing dataset. ❹ We conducted extensive experiments in four widely used datasets in MU research, demonstrating both the theoretical and practical efficacy of our approach against various state-of-the-art MU methods.

## 2 RELATED WORK

Below, we briefly review the limited existing research on MU methods and attacks against MU.

### 2.1 MACHINE UNLEARNING

**Exact Unlearning**. Retraining the model from scratch after removing specific data *(Retrain)* can intuitively and effectively achieve exact unlearning. In addition, Bourtoule et al. (2021) proposed SISA (Sharded, Isolated, Sliced, Aggregated) training, which trains isolated models on data shards for efficient unlearning by retraining only affected shards. Since forgetting data can be regarded as excluded from the training set, the success rate of unlearning can be evaluated using the Membership Inference Attack (MIA) (Shokri et al., 2017). Although effective, these unlearning approaches are computationally expensive and impractical for large-scale models and datasets.

**Approximate Unlearning.** The idea of modestly sacrificing forgetting accuracy in exchange for significant improvements in unlearning efficiency has spurred exploration of approximate unlearning techniques. Model fine-tuning *(FT)* (Warnecke et al., 2021; Golatkar et al., 2020) capitalizes on the phenomenon of catastrophic forgetting (Kirkpatrick et al., 2017), achieving unlearning by fine-tuning on the retained set of data. Gradient ascent *(GA)* (Graves et al., 2021; Golatkar et al., 2020; Thudi et al., 2022) reverses model training by adding gradients, thus moving the model towards greater loss for the data points targeted for removal. Several methods estimate the impact of removed samples on model parameters and apply modifications for efficient forgetting through the fisher information matrix *(FF)* (Becker & Liebig, 2022; Golatkar et al., 2020) or influence function *(IU)* (Koh & Liang, 2017; Izzo et al., 2021). In addition, the weight pruning *(WP)* adopted to improve the sparsity of the model could improve the effectiveness of the data erasure (Jia et al., 2023). However, residual information from unlearned data can persist in the model after approximate unlearning (Thudi et al., 2022), thus raising concerns about the ongoing risk of privacy information leakage.

### 2.2 ATTACKS ON MACHINE UNLEARNING

Despite advancements in unlearning techniques, the field faces significant challenges from various types of attacks that aim to exploit weaknesses in unlearning mechanisms. Understanding these

attacks is crucial for developing robust and secure unlearning methods. Although several attacks are proposed to affect the efficiency (Marchant et al., 2022) or fidelity (Di et al., 2022; Hu et al., 2023) of unlearning, this section will focus on data privacy attacks that are closely aligned with the objectives of this study.

**Model Inversion Attack.** Model inversion attacks aim to reconstruct the original input data from the model's outputs. Fredrikson et al. (2015) introduced model inversion attacks by leveraging confidence scores output by a model to reconstruct input images. Hu et al. (2024) proposed the first inversion attack against unlearning. It extracts features and labels of forgetting samples, which most closely match the aims of our study. Although the attack demonstrates notable effectiveness, it requires access to the original model prior to unlearning, which may be impractical. It also assumes limited scenarios, such as feature recovery with one single forgotten sample or label inference when a single category is being forgotten. In contrast, our method only requires information from the model after unlearning and supports a more versatile unlearning configuration, such as randomly forgetting multiple samples from different categories. To our knowledge, we are the first to explore the attack *solely* using unlearned models for *extensive* class membership recovery of samples, with no comparable prior work.

## 3 PRELIMINARIES

In the following sections, we first introduce the datasets and models used in our study, followed by a formal definition of the problem.

**Involved Datasets.** ∘ **Training dataset** $\mathcal{D}_t : \mathcal{X}_t \mapsto \mathcal{Y}_t$ is all the data used for initially training the machine learning models, where $\mathcal{X}_t$ denotes the image set and $\mathcal{Y}_t$ denotes the corresponding label set. ∘ **Forgetting dataset** $\mathcal{D}_f : \mathcal{X}_f \mapsto \mathcal{Y}_f$ is a subset of $\mathcal{D}_t$, i.e., $\mathcal{D}_f \in \mathcal{D}_t$. In MU, $\mathcal{D}_f$ is a set of sensitive data that should be unlearned from the trained model, i.e., the unlearned model should not tell the truth when $\mathcal{X}_f$ is input. ∘ **Remaining dataset** $\mathcal{D}_r : \mathcal{X}_r \mapsto \mathcal{Y}_r$ is the remaining data of $\mathcal{D}_t$, i.e., $\mathcal{D}_r = \mathcal{D}_t \setminus \mathcal{D}_f$, which should not be forgotten. ∘ **Probing dataset** $\mathcal{D}_p : \mathcal{X}_p \mapsto \mathcal{Y}_p$ : is a dataset constructed by the attacker. In general, $\mathcal{X}_p$ is supposed to have a similar distribution to $\mathcal{X}_f$ so that it is possible to infer $\mathcal{Y}_f$ according to $\mathcal{D}_p$.

**Involved Models.** ∘ **Trained model** $\mathcal{M}_T$ is the model that has been trained on training dataset $\mathcal{D}_t$. ∘ **Unlearned model** $\mathcal{M}_U$ is the model that has unlearned the forgetting dataset $\mathcal{D}_f$. ∘ **Attack model** $\mathcal{M}_A$ is the model that aims to recover the truth map $\mathcal{X}_f \mapsto \mathcal{Y}_f$ from the unlearned model with the probing set $\mathcal{D}_p$.

**Problem Formulation.** MU aims to remove the influence of some targeted training data $\mathcal{D}_f \in \mathcal{D}_t$ on a trained model $\mathcal{M}_T$, and release a safe unlearned model $\mathcal{M}_U$ that has forgotten the true labels $\mathcal{Y}_f$ of $\mathcal{X}_f$. This paper introduces CMIRA, a scheme specifically designed to recover sensitive data by exploiting vulnerabilities in various MU models that are supposed to forget it. To achieve this, we implement an attack model $\mathcal{M}_A$ to induce the unlearned model $\mathcal{M}_U$ to recover the class memberships of $\mathcal{X}_f$, i.e., the true labels $\mathcal{Y}_f$ of $\mathcal{X}_f$ using a probing set $\mathcal{D}_p$.

## 4 PROPOSED METHOD

In this section, we present the details of the probing dataset $\mathcal{D}_p$ construction, various MU methods addressed, and the implementation process of the proposed CMIRA scheme.

**Overview.** Figure 2 (a) Training and Unlearning illustrates the process of machine unlearning. After unlearning, the true labels of $\mathcal{X}_f$ cannot be correctly retrieved from the unlearning model $\mathcal{M}_U$. Inspired by MIA Shokri et al. (2017), we propose the CMIRA scheme which can effectively recover the true forgotten labels through the attack model $\mathcal{M}_A$ with an auxiliary dataset $\mathcal{D}_p$ to provide prior knowledge of the class memberships over $\mathcal{X}_f$. Figure 2 (b) Recovery Attack Scheme demonstrates the workflow of CMIRA. It consists of two main stages, that is, (1) *Probing Prior Learning Stage*: it trains attack model $\mathcal{M}_A$ with probing dataset $\mathcal{D}_p$, which aims to learn a class membership prior from $\mathcal{D}_p : \mathcal{X}_p \mapsto \mathcal{Y}_p$ due to the similar distributions $\mathcal{X}_p$ and $\mathcal{X}_f$; (2) *Inducing Recovery Stage*: the attack model $\mathcal{M}_A$ recovers the class memberships of $\mathcal{X}_f$, by an iterative MKD process that induces unlearned model $\mathcal{M}_U$ to output plausible labels $\hat{\mathcal{Y}}_f$.

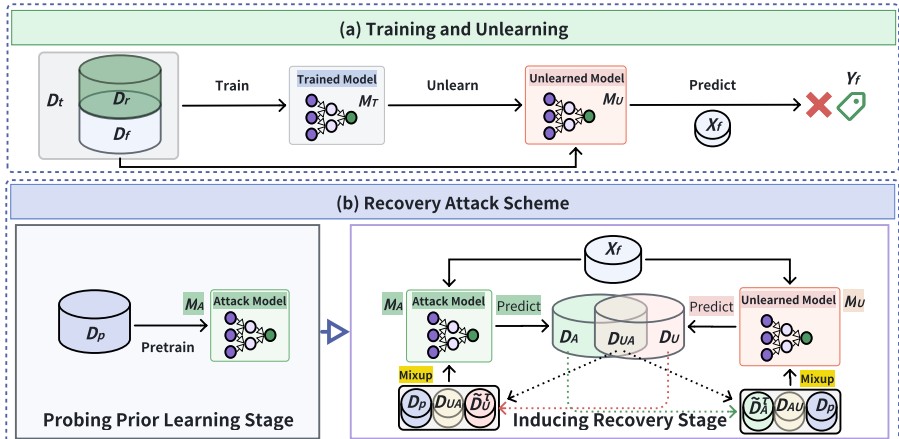

Figure 2: The framework of model training, unlearning, and recovery attack: (a) The workflow to obtain MU models; (b) The implementation of the CMIRA scheme.

**Model Training and Unlearning.** Since the proposed CMIRA scheme is agnostic to the MU model, we utilize various SOTA MU models as follows. Given the training dataset $\mathcal{D}_t = \mathcal{D}_f \cup \mathcal{D}_r$, we first use $\mathcal{D}_t$ to train a model, $\mathcal{M}_t$, with the parameters $\Theta_t$, as shown in Figure 2. Then, we apply different MU methods, as introduced in related work, on $\mathcal{M}_T$ and result in various MU models $\{\mathcal{M}_U\}$: ***RT*** is the exact MU method by retraining model parameters from scratch over the remaining dataset $\mathcal{D}_r$. ***FT*** trains model $\mathcal{M}_t$ on $\mathcal{D}_r$ using a few training epochs. The rationale of FT initiates catastrophic forgetting (Goodfellow et al., 2013). ***GA*** reverses the model training on $\mathcal{D}_f$ by moving $\Theta_t$ towards the direction of increasing loss. ***FF*** adopts additive Gaussian noise on $\Theta_t$. The noise has zero mean and covariance determined by the 4th root of the Fisher Information matrix on $\mathcal{D}_r$. ***IU*** leverages the influence function approach to characterize the change in $\Theta_t$ if a training point is removed from the training loss. ***WP*** applies binary mask $\mathbf{m}$ over model parameters $\Theta_t$, where $\mathbf{m}$ is determined by one-shot magnitude pruning (OMP) (Chen et al., 2021b) on $\mathcal{D}_f$.

## 4.1 The Recovery Attack Scheme

As shown in Figure 2 (b), the implementation of CMIRA scheme consists of two main stages: (1) Probing Prior Learning Stage and (2) Inducing Recovery Stage. We summarize the above CMIRA scheme in Algorithm 1.

**Generation of Probing Dataset.** Since both of these two stages are based on the probing dataset $\mathcal{D}_p$, we first present how to generate $\mathcal{D}_p$. In MIA (Shokri et al., 2017), the attack models are trained with shadow training data that is distributed similarly to the target model's training data. Based on the similar attacking strategy, we need to construct a probing dataset $\mathcal{D}_p$ that has a similar distribution to the forgetting dataset $\mathcal{D}_p$ to perform the class membership recovery attack. $\mathcal{D}_p$ can be easily created in the following two ways: ❶ **Image search**: The image set $\mathcal{X}_p$ is retrieved by the query set $\mathcal{X}_f$ from the image database. As a result, $\mathcal{D}_p : \mathcal{X}_p \mapsto \mathcal{Y}_p$ is constructed. Note that we may need to align the label set of the image database with the training label set. ❷ **Same data source**: If the training data is collected from some known data source, we can generate $\mathcal{D}_p$ by sampling data from this data source. In general, the datasets collected from the same data source have a similar distribution.

**Probing Prior Learning Stage.** Given the probing set $\mathcal{D}_p$ generated as the above, We can pretrain the attack model $\mathcal{M}_A$ over $\mathcal{D}_p$ to obtain the rough prior knowledge of the forgetting dataset $\mathcal{D}_f$. In particular, we implement $\mathcal{M}_A$ by placing a Lipschitz-constrained MLP with SoftMax as the output (LipSoftMax) over the backbone networks (BBN), such as the ResNet and the VGG families.

$$\mathcal{M}_A := \text{LipSoftMax}_{\mathbf{W}_L}(\text{BBN}_{SN(\mathbf{W}_B)}(x)) \tag{1}$$

$$\mathcal{P}_{M_A} := \{\mathbf{p}_x = M_A(x) | x \in \mathcal{X}_p\} \tag{2}$$

$$\Theta_A := \arg\min_{\Theta_A} L(\mathcal{P}_{M_A}, \mathcal{Y}_p) := \arg\min_{\Theta_A} \text{CrossEntropy}(\mathcal{P}_{M_A}, \mathcal{Y}_p) \tag{3}$$

where the weight matrix for each layer is adopted spectral normalization (Miyato et al., 2018), namely $SN(\mathbf{W}_L) := \mathbf{W}_L / \sigma(\mathbf{W}_L)$, to enforce the Lipschitz continuity. According to the clustering

---

**Algorithm 1** Mutual Knowledge Distillation based CMIRA Scheme

---

**Input**: Probing dataset $\mathcal{D}_p$, forgetting image set $\mathcal{X}_f$
**Output**: Recovery result $\mathcal{X}_f \mapsto \hat{\mathcal{Y}}_A$
*Probing Prior Learning Stage:*
  1: $\Theta_A := \arg\min_{\Theta_A} L(\mathcal{M}_A(\mathcal{X}_p), \mathcal{Y}_p)$          ▷ Pretrain attack model, cf Eq. (1,2,3)
*Inducing Recovery Stage:*
  1: **while** not converged **do**
  2:      $\mathcal{D}_U := (\mathcal{X}_f, \hat{\mathcal{Y}}_U, \hat{\mathcal{P}}_U), \mathcal{D}_A := (\mathcal{X}_f, \hat{\mathcal{Y}}_A, \hat{\mathcal{P}}_A)$    ▷ Construct predictive datasets, cf Eq. (4,5)
  3:      $\mathcal{D}_{UA} := \{(x, \hat{y}_U)\} \mathcal{D}_{AU} := \{(x, \hat{y}_A)\}$      ▷ Construct agreement datasets, cf Eq. (6,7)
  4:      $\bar{\mathcal{D}}_U^\tau := \{(x, \mathbf{p}_U))\}, \tilde{\mathcal{D}}_A^\tau := \{(x, \mathbf{p}_A))\}$    ▷ Construct disagreement datasets, cf Eq. (8,9)
  5:      $\tilde{\mathcal{D}}_U^{mix} := Mixup(\mathcal{D}_p \cup \mathcal{D}_{UA} \cup \bar{\mathcal{D}}_U^\tau)$    ▷ Construct mixup dataset for SSL, cf Eq. (10)
  6:      $\Theta_A \leftarrow SGD(L_{\mathcal{M}_A}(\mathcal{D}_U^{mix}), \Theta_A)$         ▷ Update $\mathcal{M}_A$ by MKD using $\mathcal{D}_U^{mix}$
  7:      $\tilde{\mathcal{D}}_A^{mix} := Mixup(\mathcal{D}_p \cup \mathcal{D}_{AU} \cup \bar{\mathcal{D}}_A^\tau)$    ▷ Construct mixup dataset for SSL, cf Eq. (10)
  8:      $\Theta_U \leftarrow SGD(L_{\mathcal{M}_U}(\mathcal{D}_A^{mix}), \Theta_U)$         ▷ Update $\mathcal{M}_U$ by MKD using $\mathcal{D}_A^{mix}$
  9: **end while**
10: **return** $\mathcal{M}_A(\Theta_A) : \mathcal{X}_f \mapsto \hat{\mathcal{Y}}_A$

---

assumption, similar inputs tend to have the same label. LipSoftMax helps to better preserve the distance distribution between the features of images output by the BBN and its corresponding label embedding vector. Consequently, LipSoftMax can better infer the forget label $\mathcal{Y}_f$ according to $\mathcal{Y}_p$ by exploiting the similar distributions between $\mathcal{X}_f$ and $\mathcal{X}_p$

**Inducing Recovery Stage.** Although each MU model $\mathcal{M}_U$ has performed unlearning on $\mathcal{D}_f$, it still retains the classification capability on the remaining dataset $\mathcal{D}_r$. Since both $\mathcal{D}_f$ and $\mathcal{D}_r$ belong to the training dataset $\mathcal{D}_t$, it is possible to induce $\mathcal{M}_U$ to recover the class membership of $\mathcal{X}_f$. In this paper, we design an iterative MKD process to transfer knowledge between the unlearning model $\mathcal{M}_U$ and the attack model $\mathcal{M}_A$. First, we collect the predictive labels on $\mathcal{X}_f$ as follows where We denote the predictive datasets from $\mathcal{M}_U$ and $\mathcal{M}_A$ as $\mathcal{D}_U$ and $\mathcal{D}_A$:

$$\mathcal{D}_U := (\mathcal{X}_f, \hat{\mathcal{Y}}_U), \hat{\mathcal{P}}_U \quad \text{i.e. } \mathcal{P}_U := \mathcal{M}_U(\mathcal{X}_f), \ \hat{\mathcal{Y}}_U := \{\arg\max_c \mathbf{p}(c) | \mathbf{p} \in \mathcal{P}_U\} \tag{4}$$

$$\mathcal{D}_A := (\mathcal{X}_f, \hat{\mathcal{Y}}_A), \hat{\mathcal{P}}_A \quad \text{i.e. } \mathcal{P}_A := \mathcal{M}_A(\mathcal{X}_f), \ \hat{\mathcal{Y}}_A := \{\arg\max_c \mathbf{p}(c) | \mathbf{p} \in \mathcal{P}_A\} \tag{5}$$

Then, we can easily extract the agreement subsets $\mathcal{D}_{UA}$ and $\mathcal{D}_{AU}$ in terms of the consistent predictive labels $\hat{\mathcal{Y}}_U$ and $\hat{\mathcal{Y}}_A$ output by $\mathcal{M}_U$ and $\mathcal{M}_A$:

$$\mathcal{D}_{UA} := \{(x, \hat{y}_U), \mathbf{p}_U \mid \hat{y}_U(x) = \hat{y}_A(x); x \in \mathcal{X}_f\} \tag{6}$$

$$\mathcal{D}_{AU} := \{(x, \hat{y}_A), \mathbf{p}_A \mid \hat{y}_U(x) = \hat{y}_A(x); x \in \mathcal{X}_f\} \tag{7}$$

Correspondingly, we can denote the disagreement subsets $\bar{\mathcal{D}}_{UA} := \mathcal{D}_U \setminus \mathcal{D}_{UA}$ and $\bar{\mathcal{D}}_{AU} := \mathcal{D}_A \setminus \mathcal{D}_{AU}$. Then we extract, respectively, a small proportion of data from $\bar{\mathcal{D}}_U$ and $\bar{\mathcal{D}}_A$ with the highest predictive confidence above threshold $\tau$:

$$\tilde{\mathcal{D}}_U^\tau := \{(x, \mathbf{p}_U) \mid \max_c \mathbf{p}(c) \geq \tau, (x, y_U), \mathbf{p} \in \bar{\mathcal{D}}_{UA}\} \tag{8}$$

$$\tilde{\mathcal{D}}_A^\tau := \{(x, \mathbf{p}_A) \mid \max_c \mathbf{p}(c) \geq \tau, (x, y_A), \mathbf{p} \in \bar{\mathcal{D}}_{AU}\} \tag{9}$$

Given a set of probing images $\mathcal{X}_p$ with corresponding ground-truth labels $\mathcal{Y}_p$, and a set of forgetting images $\mathcal{X}_f$ with unknown labels, the problem can be naturally formulated within the framework of semi-supervised learning (SSL). Recent work (Berthelot et al., 2019) has shown that Mixup (Hongyi Zhang, 2018), a simple yet highly effective data augmentation technique, can lead to substantial improvements in SSL performance.

$$\tilde{\mathcal{D}} := \{\tilde{x}, \tilde{\mathbf{p}}\} := Mixup\big((x_1, \mathbf{p}_1), (x_2, \mathbf{p}_2)\big) \ for \ (x_1, \mathbf{p}_1), (x_2, \mathbf{p}_2) \sim \mathcal{D} \tag{10}$$

$$\text{where} \quad \tilde{x} = \lambda x_1 + (1 - \lambda)x_2, \ \tilde{\mathbf{p}} = \lambda \mathbf{p}_1 + (1 - \lambda)\mathbf{p}_2, \ \lambda \sim Beta(\alpha, \alpha) \tag{11}$$

Table 1: The overall evaluation of CMIRA's attack efficacy. All the metric scores are reported by (%)

| Dataset | Metric | ResNet18 | | | | | | VGG16 | | | | |
|---|---|---|---|---|---|---|---|---|---|---|---|---|
| | | RT | FT | FF | GA | IU | WP | RT | FT | GA | IU | WP |
| Cifar-10 | $\mathbf{Acc}_U$ | 66.25 | 66.87 | 51.82 | 76.34 | 87.47 | 40.79 | 77.66 | 79.80 | 59.38 | 52.73 | 61.90 |
| | $\mathbf{Acc}_A$ | 69.40 | 70.17 | 95.66 | 92.13 | 97.08 | 54.01 | 80.77 | 82.60 | 82.38 | 81.18 | 70.80 |
| | $\mathbf{R}_R$ | 4.76 | 4.93 | 84.60 | 20.69 | 10.99 | 32.40 | 4.01 | 3.50 | 38.74 | 53.96 | 14.37 |
| Cifar-100 | $\mathbf{Acc}_U$ | 19.56 | 22.58 | 50.76 | 29.16 | 55.20 | 15.73 | 28.80 | 34.67 | 14.40 | 38.93 | 20.18 |
| | $\mathbf{Acc}_A$ | 21.96 | 22.49 | 96.89 | 93.69 | 96.18 | 16.62 | 32.62 | 38.04 | 89.78 | 94.76 | 26.84 |
| | $\mathbf{R}_R$ | 12.27 | -0.39 | 90.89 | 221.34 | 74.24 | 5.65 | 13.27 | 9.74 | 523.46 | 143.38 | 33.04 |
| TinyImg | $\mathbf{Acc}_U$ | 29.20 | 31.52 | 59.68 | 44.56 | 14.96 | 31.04 | 38.72 | 41.04 | 34.88 | 23.04 | 36.48 |
| | $\mathbf{Acc}_A$ | 38.96 | 39.52 | 96.88 | 98.56 | 98.64 | 52.80 | 53.92 | 47.92 | 99.04 | 98.48 | 51.76 |
| | $\mathbf{R}_R$ | 33.42 | 25.38 | 62.33 | 121.18 | 559.36 | 70.10 | 39.26 | 16.76 | 183.94 | 327.43 | 41.89 |
| FMNIST | $\mathbf{Acc}_U$ | 95.82 | 96.23 | 74.02 | 77.33 | 46.16 | 24.39 | 96.15 | 96.94 | 27.70 | 27.43 | 92.90 |
| | $\mathbf{Acc}_A$ | 96.22 | 96.83 | 89.88 | 96.55 | 81.25 | 35.56 | 96.63 | 97.39 | 49.61 | 75.45 | 94.61 |
| | $\mathbf{R}_R$ | 0.42 | 0.62 | 21.42 | 24.85 | 76.01 | 45.81 | 0.50 | 0.47 | 79.14 | 175.07 | 1.84 |

**Key Insights.** In Algorithm 1, the probing dataset $\mathcal{D}_p$ is used to optimize both $\mathcal{M}_U$ and $\mathcal{M}_A$ to learn the map $\mathcal{X}_p \mapsto \mathcal{Y}_p$ that provides the strongest prior to better inferring $\mathcal{X}_f \mapsto \mathcal{Y}_f$. The subsets $\mathcal{D}_{UA}$ and $\mathcal{D}_{AU}$ align $\mathcal{M}_U$ and $\mathcal{M}_A$ with consistently aligned labels: $\hat{y}_U$ and $\hat{y}_A$ which helps recover the original class membership based on mutual agreement. Moreover, $\tilde{\mathcal{D}}_U^\tau$ and $\tilde{\mathcal{D}}_A^\tau$ are the disagreement subsets with high confidence, which performs MKD to align $\mathcal{M}_A$ and $\mathcal{M}_U$ Han et al. (2018). The above step is performed iteratively to build agreement between $\mathcal{M}_A$ and $\mathcal{M}_U$ as much as possible. Finally, we obtain the result of recovered class memberships, $\mathcal{M}_A(\Theta_A) : \mathcal{X}_f \mapsto \hat{\mathcal{Y}}_A$.

## 5 EXPERIMENTS

### 5.1 EXPERIMENT SETUP

**Datasets.** Four widely used image classification datasets are used in our experiments. These include **Cifar-10** and **Cifar-100** (Krizhevsky et al., 2009), which consists of 32x32 color images with 10 and 100 classes respectively; Tiny-ImageNet-200 (**TinyImg** (Le & Yang, 2015), which contains 200 classes of 64x64 color images; and Fashion-MNIST (**FMNIST** (Xiao et al., 2017), a dataset featuring 28x28 grayscale images of 10 different apparel items.

**Target Machine Unlearning Models.** Since CMIRA is an MU model-agnostic attack scheme, we comprehensively evaluated the performance of CMIRA against six SOTA MU models as described in *Proposed Method*, including Retrain (**RT**), Fine-Tune (**FT**), Gradient Ascend (**GA**), Fish Forgetting (**FF**), Influence Unlearning (**IU**), and Weight Prune (**WP**). Moreover, we further assessed the performance of two representative backbone architectures for image classification: **ResNet18** (He et al., 2016) and **VGG16** (Simonyan & Zisserman, 2014). These two model architectures are widely used in the evaluation of SOTA MU methods. We first trained the RestNet18 and VGG16 models over the training set $\mathcal{D}_t$ for each experimental dataset, i.e., Cifar-10, Cifar-100, TinyImage and FMNIST. Then, each of the above MU methods was performed on the trained models to obtain the various unlearned models. This diversity of MU models over different datasets and backbone architectures allows for a comprehensive assessment of CMIRA's efficacy in various MU scenarios.

**Experimental Details.** The official dataset splits (e.g., Cifar-10: 80% for training, 20% for testing) are used for the evaluation of MU methods. The training set is used as $\mathcal{D}_t$ to train the backbone models. Five classes from $\mathcal{D}_t$ are selected to construct the forgetting dataset $\mathcal{D}_f$ by randomly sampling 50% of data from them. In general, the split set for testing is similarly distributed to $\mathcal{D}_f$ since they were collected from the same data source. Therefore, the testing split is a suitable data source to generate the probing dataset $\mathcal{D}_p$ as described in *Proposed Method*. All experiments utilized an SGD optimizer and were conducted on 8 NVIDIA A100 GPUs. More details of the experiments can be found in the supplementary materials.

### 5.2 EVALUATION METRICS

The effectiveness of recovery attacks can be intuitively assessed by evaluating the prediction accuracy (**Acc**) on the set of forgetting inputs $\mathcal{X}_f$. To provide a more comprehensive and detailed evaluation of

Table 2: The evaluation results on efficacy of class membership recovery. In each cell, we report $\mathcal{M}_U$'s prediction accuracy $\mathbf{Acc}_U$ in percentage (%) with its corresponding recovery improvement $\uparrow \mathbf{R}_I$ achieved by CMIRA.

| Model | Method | Class #1 Airplane | Class #2 Automobile | Class #3 Cat | Class #4 Dog | Class #5 Frog | $\mathbf{A}_R$ (%) |
|-------|--------|---------|------------|---------|---------|---------|---------|
| ResNet18 | RT | 70.71 ↑ 5.51 | 79.07 ↑ 2.66 | 48.58 ↑ 2.35 | 57.11 ↑ 2.40 | 75.78 ↑ 2.84 | 11.20 |
| | FT | 72.36 ↑ 4.13 | 80.58 ↑ 2.71 | 49.38 ↑ 3.55 | 55.96 ↑ 0.97 | 76.09 ↑ 5.11 | 10.54 |
| | FF | 34.49 ↑ 56.58 | 99.91 ↑ 0.05 | 33.42 ↑ 60.00 | 33.16 ↑ 62.97 | 58.13 ↑ 39.60 | 314.98 |
| | GA | 78.80 ↑ 14.44 | 75.64 ↑ 16.49 | 73.29 ↑ 18.58 | 77.16 ↑ 14.08 | 76.80 ↑ 15.38 | 44.68 |
| | IU | 94.84 ↑ 3.60 | 86.76 ↑ 9.55 | 85.82 ↑ 11.02 | 86.62 ↑ 10.85 | 83.29 ↑ 13.02 | 20.31 |
| | WP | 43.42 ↑ 20.45 | 51.69 ↑ 14.80 | 36.49 ↑ 1.95 | 24.84 ↑ 9.96 | 47.51 ↑ 18.93 | 84.69 |
| VGG16 | RT | 81.07 ↑ 4.17 | 89.24 ↑ 1.52 | 65.96 ↑ 4.26 | 66.80 ↑ 3.51 | 85.24 ↑ 2.09 | 8.71 |
| | FT | 81.91 ↑ 4.49 | 90.31 ↑ 0.98 | 69.56 ↑ 3.86 | 71.60 ↑ 2.27 | 85.64 ↑ 2.36 | 7.96 |
| | GA | 64.36 ↑ 23.15 | 38.31 ↑ 23.29 | 61.07 ↑ 25.86 | 60.22 ↑ 24.00 | 72.93 ↑ 18.71 | 90.99 |
| | IU | 62.71 ↑ 25.11 | 38.67 ↑ 24.97 | 55.64 ↑ 31.47 | 41.78 ↑ 35.11 | 64.84 ↑ 25.60 | 129.66 |
| | WP | 72.71 ↑ 8.36 | 85.91 ↑ 4.22 | 33.60 ↑ 11.20 | 61.96 ↑ 2.13 | 55.33 ↑ 18.58 | 31.57 |

the recovery attack's effectiveness, we introduced two additional metrics: ❶ **Recovery Rate**($\mathbf{R}_I$) and ❷ **Area of Membership Recovery**($\mathbf{A_R}$), both of which are briefly described below.

**Recovery Rate ($\mathbf{R}_R$)** is defined as the relative improvement of prediction accuracy: $\mathbf{R}_R = \frac{\mathbf{Acc}_A - \mathbf{Acc}_U}{\mathbf{Acc}_U}$, where $\mathbf{Acc}_A$ represents the accuracy of the attack model $\mathcal{M}_A$ on the forgetting set $\mathcal{X}_f$, and $\mathbf{Acc}_U$ is the accuracy of the unlearned model $\mathcal{M}_U$. To simplify this expression, we define the numerator as the **Recovery Improvement ($\mathbf{R}_I$)**: $\mathbf{R}_I = \mathbf{Acc}_A - \mathbf{Acc}_U$.

**Area of Membership Recovery ($\mathbf{A_R}$)** evaluates the recovery capability of $\mathcal{M}_A$ from multi-class perspective. First, we can calculate the accuracy for each forgetting class, which forms a polygon as shown in Figure 4. Then, we calculate the area of the polygon and get the value of area $\mathcal{A}_A$ for $\mathcal{M}_A$ and $\mathcal{A}_U$ for $\mathcal{M}_U$. Accordingly, $\mathbf{A_R}$ is defined as: $\mathbf{A}_R = \frac{\mathcal{A}_A - \mathcal{A}_U}{\mathcal{A}_U}$.

For further details, please refer to the supplementary material.

## 5.3 MAIN RESULTS

### 5.3.1 OVERALL EFFICACY OF CMIRA

We comprehensively evaluated the recovery attack performance of CMIRA across four datasets and diverse configurations of the MU model, as reported in Table 1. From all of the results, it is easy to find that the proposed CMIRA scheme is capable of significantly improving the prediction accuracy on the forgetting data for all MU models targeted, which strongly proves that CMIRA is a very effective and versatile MU model-agnostic method to recover the forgetting data. However, a closer look at Table 1 reveals that the recovery rate $\mathbf{R}_R$ for **RT** and **FT** is relatively lower than for other MU methods. We attribute this to **RT** is the exact unlearning method that is not trained with any forgetting data, i.e., no knowledge can be transferred to the attack model. Similarly, **FT** is able to reach a similar effect to **RT** due to catastrophic forgetting. In comparison, other MU methods retain

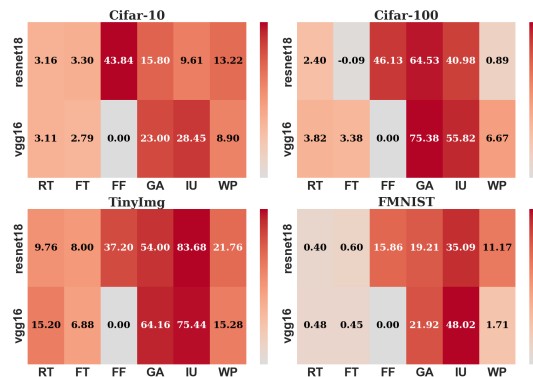

Figure 3: The recovery improvement $\mathbf{R}_I$ by CMIRA is reported in percentage (%). Each subplot displays the results for a specific dataset, with the horizontal axis representing the various MU methods and the vertical axis representing different backbone architectures.

more forgetting data-related knowledge inside the models, so CMIRA can effectively induce the remaining knowledge from these models to achieve a high recovery rate.

As illustrated in Figure 3, the absolute improvement in prediction accuracy by CMIRA is evident from a more intuitive perspective. This robust performance highlights the potential of CMIRA as a valuable tool in assessing the risk of privacy information leakage associated with MU methods, thereby facilitating the development of more effective and robust approaches.

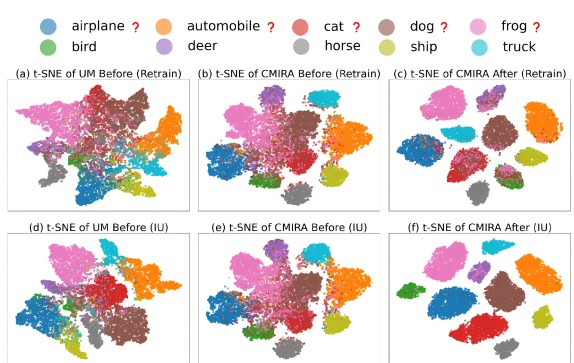

Figure 4: Class membership recovery polygon over five forgetting classes. The red area represents the prediction accuracy of GA model for each class, while the blue area represents the prediction accuracy of CMIRA. '#$n$' represents the $n$-th forgetting class.

### 5.3.2 Efficacy of Class Membership Recovery

As implied by the name CMIRA, our primary objective is to recover the true class memberships for the forgetting images $\mathcal{X}_f$. We further conducted in-depth evaluations w.r.t. each forgetting class over all four datasets and MU models. Table 2 demonstrates the performance on Cifar-10. By carefully checking each cell of the table, we can find the recovery improvement $\mathbf{R}_I$ is consistently positive across all classes. Moreover, we further visualize the results with radar charts in Figure 4. It is easy to find that the accuracy polygons of CMIRA envelop those of MU models with obvious margins for all cases, and *AMR* is reported in the last column of Table 2. Through these comprehensive evaluations, it can be concluded that CMIRA is an effective approach to the recovery of forgetting class memberships. Due to the space limits, more results can be found in the supplement.

### 5.3.3 Class membership Visualization

Figure 5 demonstrates the t-SNE of Cifar-10 datapoints w.r.t. unlearned models, pretrained attack models, and CMIRA models. The forgetting data are heavily mixed with the remaining data where no clear class boundaries can be found.

The t-SNE of pretrained attack models shows some improvement in forgetting data thanks to the knowledge learned from the probing data $\mathcal{D}_p$, but most classes are still mixed together. In comparison, the t-SNE of CMIRA shows clear boundaries between classes, and most samples are correctly assigned to their labeled clusters. We attribute this to the effectiveness of the proposed CMIRA approach in recovering forgotten class memberships.

Figure 5: t-SNE plots of Cifar-10 datapoints in $\mathcal{D}_f$ and $\mathcal{D}_p$ w.r.t. unlearned models, pretrained attack models, and CMIRA models. The legend labels followed by a question mark indicate the forgetting classes.

### 5.4 Ablation Study

In this section, we discuss the effectiveness of each component in the implementation of CMIRA framework. The models for the ablation study include: ❶ $\mathcal{M}^P$: this attack model is obtained by only performing the pretraining over probing set $\mathcal{D}_p$, i.e. 1st stage only. ❷ $\mathcal{M}^{P+U}$: this attack model is obtained by freezing the MU model and only updating the attack model, i.e. single-direction knowledge distillation. ❸ $\mathcal{M}^{P+U+A}$: the full attack model presented in this paper with pretraining and MKD.

Table 3: Results of ablative models are evaluated on Cifar-10 with the backbone of ResNet18. The accuracy of MU models $\mathbf{Acc}_U$ in percentage (%) is reported as baseline, P, P+U and P+U+A stand for the models $\mathcal{M}^P$, $\mathcal{M}^{P+U}$, and $\mathcal{M}^{P+U+A}$.

| MU Method | RT | FT | FF | GA | IU | WP |
|---|---|---|---|---|---|---|
| Baseline | 66.25 | 66.87 | 51.82 | 76.34 | 87.47 | 40.79 |
| P | 57.49 | 57.49 | 57.49 | 57.49 | 57.49 | 57.49 |
| P+U | 66.27 | 67.05 | 53.49 | 76.29 | 87.45 | 47.15 |
| **P+U+A** | **69.40** | **70.17** | **95.66** | **92.13** | **97.08** | 54.01 |

### 5.4.1 Comparison Results

Table 3 reports the results obtained on Cifar-10 using ResNet18. The precision of each MU model on forgetting dataset $\mathcal{D}_f$ is reported as the baseline. Please refer to the supplementary material for additional results on other datasets and model architectures. From the results, the full model

$\mathcal{M}^{P+U+A}$ overall outperforms the ablative models $\mathcal{M}^P$ and $\mathcal{M}^{P+U}$. Note that $\mathcal{M}^P$ is simply trained on $\mathcal{D}_p$ irrelevant to any MU models, so the results are identical across different MU methods. The low performance of $\mathcal{M}^P$ can be attributed to the difference of distributions between the probing data and the forgetting data in nature. As a result, even below the baseline, such as in **RT**, **FT**, **GA**, and **IU** scenarios. $\mathcal{M}^{P+U}$ outperforms $\mathcal{M}^P$ due to the one-way distillation of knowledge from the MU models to the attack model. However, MU models can only transfer the unforgotten knowledge to the attack model, whereas the information on forgetting data is very limited or even wrong. As a result, the recovery rate of $\mathcal{M}^{P+U}$ is accordingly small. Through the iterative distillation of mutual knowledge between the attack model and the MU model, both models can keep improving their classification upper bound by utilizing the co-agreement and disagreement knowledge (see Algorithm 1). As a result, the full model $\mathcal{M}^{P+U+A}$ achieves the best recovery rate.

### 5.4.2 VISUALIZATION OF CONFUSION MATRIX

Figure 6 (a-c) presents the normalized confusion matrices of $\mathcal{D}_f$ with respect to $\mathcal{M}^P$, $\mathcal{M}^{P+U}$, and $\mathcal{M}^{P+U+A}$. From these visualizations, we observe consistent patterns that align with the results reported in Table 3.

In the case of $\mathcal{M}^P$, we observe a relatively high prediction error rate, particularly in the five forgetting classes. The confusion matrix shows significant off-diagonal elements, suggesting a substantial misclassification due to the distribution difference between $\mathcal{D}_p$ and $\mathcal{D}_f$. Although the non-forgetting classes maintain better diagonal accuracy, there is still slight performance degradation, highlighting the challenge of generalization.

For $\mathcal{M}^{P+U}$, a marginal improvement is observed. The confusion matrix reveals a somewhat clearer diagonal, implying that the additional information from $\mathcal{D}_f$ helps retain some

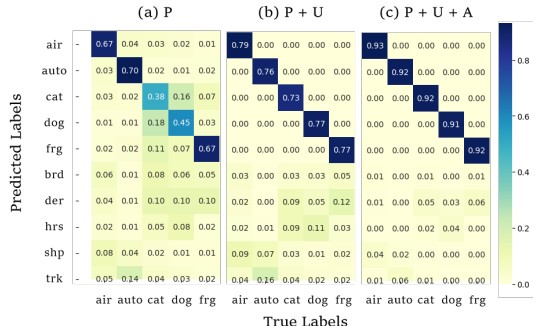

Figure 6: The plots of normalized confusion matrices demonstrate the classification performance of ablative models $\mathcal{M}^P$, $\mathcal{M}^{P+U}$, and $\mathcal{M}^{P+U+A}$ on Cifar-10 using the GA method. The labels are reordered (the five forgetting classes are listed first) to better emphasize the class membership recovery capability achieved by CMIRA with the MKD technique.

class knowledge. However, this recovery is limited as off-diagonal misclassifications remain significant, especially in forgetting classes. Nevertheless, the improvement in non-forgetting classes suggests that the model benefits from the residual information, though it is not yet sufficient for fully restoring forgotten class memberships.

The most significant improvement comes with $\mathcal{M}^{P+U+A}$, where the confusion matrix exhibits a sharp diagonal, particularly in the forgetting classes. This model achieves near-perfect classification in these classes, indicating the success of the MKD technique in restoring true class memberships. Moreover, $\mathcal{M}^{P+U+A}$ manages to balance performance across both forgetting and non-forgetting classes, without sacrificing accuracy in either set. This result underscores the importance of using auxiliary information along with a robust distillation mechanism to effectively mitigate forgetting.

## 6 CONCLUSION

In this study, we present Class Membership Inducing Recovery Attack (CMIRA), a novel attack method that can recover true class memberships from machine unlearning (MU) models without needing access to the original model. By using mutual knowledge distillation (MKD) with a probing dataset, CMIRA effectively retrieves forgotten labels. Our experiments with four widely used datasets show that CMIRA is both theoretically sound and practically effective against various MU methods. Our findings highlight the need for future research to focus on developing more robust MU systems and establish new benchmarks for evaluating their security.

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

# Appendix

## A DETAILED EXPERIMENTAL SETUP

### A.1 DATASETS

This part provides an expanded interpretation of the **Datasets** section within the **Experiment Setup** part of the main paper.

For each dataset, we used the official training and testing splits, such as 50,000 images for the training and 10,000 for the testing in Cifar-10 (Krizhevsky et al., 2009) 90 % of the training set is used as $\mathcal{D}_t$ to train the initial model $\mathcal{M}_T$ with the remaining 10% for validation, while the probing dataset $\mathcal{D}_p$ was constructed from the testing set for pretraining the recovery attack model $\mathcal{M}_A$. Subsequently, we selected data from five categories within $\mathcal{D}_t$, using half of the data from each selected category to form the forgetting dataset $\mathcal{D}_f$, and the rest serving as $\mathcal{D}_r$. Various MU methods were then applied to $\mathcal{M}_T$ to produce the unlearned models $\mathcal{M}_U$ with $\mathcal{D}_f$ and $\mathcal{D}_r$. The detailed settings for each data set are shown in Table A1.

Table A1: Details of the dataset split. For TinyImg we used 100% of training set as $\mathcal{D}_t$ because it provides additional validation set with 10,000 images.

| Dataset | Train | Test | $\mathcal{D}_t$ | $\mathcal{D}_f$ | $\mathcal{D}_p$ |
|---------|-------|------|------|------|------|
| Cifar-10 | 50,000 | 10,000 | 45,000 | 2250×5 | 10,000 |
| Cifar-100 | 50,000 | 10,000 | 45,000 | 225×5 | 10,000 |
| TinyImg | 100,000 | 10,000 | 100,000* | 250×5 | 10,000 |
| FMNIST | 60,000 | 10,000 | 54,000 | 2700×5 | 10,000 |

Table A2: Detailed parameters for training machine unlearning models across various datasets and backbone networks. The table presents the parameters used for each dataset: CIFAR-10, CIFAR-100, TinyImageNet, and FMNIST. **FC** refers to the forgotten classes through machine unlearning. Specifically, for CIFAR-10, the forgotten classes 0, 1, 3, 5, and 6 correspond to Airplane, Automobile, Cat, Dog, and Frog, respectively. For CIFAR-100, classes 11, 22, 33, 44, and 55 represent Boy, Clock, Forest, Lizard, and Otter. For TinyImageNet, classes 1, 51, 101, 151, and 198 refer to Salamander, Baboon, Hammer, Umbrella, and Slug. Lastly, for FMNIST, classes 1, 3, 5, 7, and 9 stand for Trouser, Dress, Sandal, Sneaker, and Ankle Boot. $N_{\textbf{FC}}$ indicates the number of forgotten samples in each class. Epochs refers to the number of epochs used for training the machine unlearning models. **Unlearn_lr** represents the learning rate applied during the training of the machine unlearning models. **Alpha** denotes the scaling hyperparameter on updating model parameters during the training of the machine unlearning models. This comprehensive summary provides the details of reproducing and understanding of the machine unlearning processes applied in this study.

| Dataset | #Para | ResNet18 | | | | | | VGG16 | | | | | |
|---------|-------|------|------|------|------|------|------|------|------|------|------|------|------|
| | | RT | FT | GA | FF | IU | WP | RT | FT | GA | FF | IU | WP |
| Cifar-10 | FC | | | Airplane (#0) | | Automobile (#1) | | Cat (#3) | Dog (#5) | Frog (#6) | | | |
| | $N_{\text{FC}}$ | 2250 | 2250 | 2250 | 2250 | 2250 | 2250 | 2250 | 2250 | 2250 | 2250 | 2250 | 2250 |
| | Epochs | 100 | 100 | 4 | 100 | 100 | 50 | 100 | 100 | 4 | 100 | 100 | 50 |
| | Unlearn_lr | 0.1 | 0.1 | 0.0001 | 0.1 | 0.1 | 0.01 | 0.1 | 0.1 | 0.0001 | 0.1 | 0.1 | 0.01 |
| | Alpha | NA | NA | NA | 16.5 | 16 | 0.005 | NA | NA | NA | 16.5 | 40 | 0.005 |
| Cifar-100 | FC | | | Boy (#11) | | Clock (#22) | | Forest (#33) | Lizard (#44) | Otter (#55) | | | |
| | $N_{\text{FC}}$ | 225 | 225 | 225 | 225 | 225 | 225 | 225 | 225 | 225 | 225 | 225 | 225 |
| | Epochs | 100 | 100 | 4 | 100 | 100 | 50 | 100 | 100 | 4 | 100 | 100 | 50 |
| | Unlearn_lr | 0.1 | 0.1 | 0.001 | 0.1 | 0.1 | 0.001 | 0.1 | 0.1 | 0.001 | 0.1 | 0.1 | 0.001 |
| | Alpha | NA | NA | NA | 20 | 160 | 0.005 | NA | NA | NA | 16.5 | 200 | 0.005 |
| TinyImg | FC | | | Salamander (#11) | | Baboon (#51) | | Hammer (#101) | Umbrella (#151) | Slug (#198) | | | |
| | $N_{\text{FC}}$ | 250 | 250 | 250 | 250 | 250 | 250 | 250 | 250 | 250 | 250 | 250 | 250 |
| | Epochs | 100 | 100 | 5 | 100 | 100 | 50 | 100 | 100 | 4 | 100 | 100 | 50 |
| | Unlearn_lr | 0.1 | 0.1 | 0.00001 | 0.1 | 0.1 | 0.001 | 0.1 | 0.1 | 0.0001 | 0.1 | 0.1 | 0.002 |
| | Alpha | NA | NA | NA | 20 | 160 | 0.001 | NA | NA | NA | 16.5 | 100 | 0.001 |
| FMNIST | FC | | | Trouser (#1) | | Dress (#3) | | Sandal (#5) | Sneaker (#7) | Ankle Boot (#9) | | | |
| | $N_{\text{FC}}$ | 250 | 250 | 250 | 250 | 250 | 250 | 250 | 250 | 250 | 250 | 250 | 250 |
| | $N_{\text{FC}}$ | 2700 | 2700 | 2700 | 2700 | 2700 | 2700 | 2700 | 2700 | 2700 | 2700 | 2700 | 2700 |
| | Epochs | 100 | 100 | 5 | 100 | 100 | 50 | 100 | 100 | 4 | 100 | 100 | 50 |
| | Unlearn_lr | 0.1 | 0.1 | 0.00001 | 0.1 | 0.1 | 0.02 | 0.1 | 0.1 | 0.0001 | 0.1 | 0.1 | 0.005 |
| | Alpha | NA | NA | NA | 16.5 | 100 | 0.03 | NA | NA | NA | 16.5 | 40 | 0.02 |

## A.2 TRAINING DETAILS

This part provides an expanded interpretation of the **Experimental Details** section within the **Experiment Setup** part in the main body of the paper.

In our experiment, we used the SGD optimizer for training. $\mathcal{M}_T$ was trained for 100 epochs with $lr = 0.1$. During training of $\mathcal{M}_U$, we adopted the hyper-parameter settings recommended in (Jia et al., 2023) for various unlearning methods and made fine adjustments on this basis, we can find detailed parameters for training $\mathcal{M}_U$ in Table A2. In implementing CMIRA, we first pre-trained $\mathcal{M}_A$ for 200 epochs with $lr = 0.01$. In the inducing recovery stage, we iteratively trained both $\mathcal{M}_A$ and $\mathcal{M}_U$, and updated $\mathcal{D}_{UA}$ over 200 iterations. In each iteration, $\mathcal{M}_A$ and $\mathcal{M}_U$ were trained for 10 epochs with $lr = 0.001$. We employed an early-stop strategy, terminating the process if the recovery accuracy did not improve for 7 consecutive iterations. All experiments were conducted on the computing system equipped with 8 NVIDIA® A100 GPUs.

## B  DETAILED EVALUATION METRICS

This part provides an expanded interpretation of the **Evaluation metrics** section within the **Experiments** part in the main body of the article. Below we explain the evaluation metrics we used in our study in more detail.

### B.1  RECOVERY RATE ($\mathbf{R}_R$)

It is used to assess the comprehensive recovery capability of CMIRA for forgotten data. We denote the accuracy of the attack model $\mathcal{M}_A$ in forgetting data $\mathcal{X}_f$ as $\mathbf{Acc}_A$ and that of the unlearned model $\mathcal{M}_U$ as $\mathbf{Acc}_U$, and the difference between them can reflect the extent of recovery in class membership prediction, which could be defined as **Recovery Improvement ($\mathbf{R}_I$)**:

$$\mathbf{R}_I = \mathbf{Acc}_A - \mathbf{Acc}_U \tag{12}$$

Due to significant variations in $\mathbf{Acc}_U$ across different model and unlearn method configurations, we focus on its relative recovery rate $\mathbf{R}_R$, defined as:

$$\mathbf{R}_R = \frac{\mathbf{R}_I}{\mathbf{Acc}_U} = \frac{\mathbf{Acc}_A - \mathbf{Acc}_U}{\mathbf{Acc}_U} \tag{13}$$

### B.2  AREA OF MEMBERSHIP RECOVERY ($\mathbf{A}_R$)

It evaluates the recovery capability of CMIRA from a multi-class perspective. We define the concept of the Membership Recovery Polygon (MRP) to facilitate the evaluation. The polygon is generated through the radar chart, where each vertex of the polygon in the radar chart corresponds to a specific class, with the distance from the center to the point indicating the accuracy level of that class (e.g. $\mathbf{Acc}_U^i$ or $\mathbf{Acc}_A^i$ for the $i$-th class). By plotting these points and connecting them sequentially, the Membership Recovery polygon (MRP) is formed. This graphical representation provides an intuitive overview of performance in different classes, highlighting recovery effectiveness in a comparative context.

We further obtain the area of the polygon, $\mathcal{A}_A$ for $\mathcal{M}_A$ and $\mathcal{A}_U$ for $\mathcal{M}_U$:

- $\mathcal{A}_U$: This metric indicates that the smaller it is, the better the unlearning effect of $\mathcal{M}_U$, meaning the poorer the membership memory retention, which is the goal of various unlearning methods.

- $\mathcal{A}_A$: This metric indicates that the larger it is, the better the recovery effect of $\mathcal{M}_A$, meaning the better the membership recovery after the attack.

Similar to $\mathbf{R}_R$, we prioritize its relative recovery rate $\mathbf{A}_R$, defined as:

$$\mathbf{A}_R = \frac{\mathcal{A}_A - \mathcal{A}_U}{\mathcal{A}_U} \tag{14}$$

## C  ADDITIONAL MAIN RESULTS

This section provides an expanded interpretation of the **Main Results Details** section within the **Experiments** part in the main body of the paper.

### C.1  CLASS MEMBERSHIP RECOVERY

This part provides an expanded interpretation of the **Efficacy of Class Membership Recovery** section within the **Main Results** part in the main body of the paper.

We evaluated our proposed method, CMIRA, on four datasets—CIFAR-10, CIFAR-100 (Krizhevsky et al., 2009), TinyImageNet (Le & Yang, 2015), and FMNIST (Xiao et al., 2017)—using two backbone networks, ResNet18 (He et al., 2016) and VGG16 (Simonyan & Zisserman, 2014). We tested its performance across six different MU methods: Retrain, FF (Becker & Liebig, 2022; Golatkar et al., 2020), FT (Warnecke et al., 2021; Golatkar et al., 2020), GA (Graves et al., 2021; Golatkar et al., 2020; Thudi et al., 2022), IU (Koh & Liang, 2017; Izzo et al., 2021), and WP (Jia et al., 2023). As shown in Table A3 and Figure A1, applying the CMIRA attack strategy to models trained on these datasets for predicting forgotten classes led to significant improvements both in $\mathbf{R}_I^i$, which is the recovery improvement of each class $i$ and in the area of membership recovery $\mathbf{A}_R$. These metrics showed substantial enhancements compared to the performance of the original MU models.

### C.2  T-SNE VISUALIZATION

This part provides an expanded interpretation of the **Class membership Visualization** section within the **Main Results** part in the main body of the paper.

Due to space limitations, the main paper presents the t-SNE plots for the Retrain and IU methods, while the supplementary material encompasses the t-SNE plots corresponding to the four remaining MU methods (FF, FT, WP, and GA). The visualization results of Figure A2 demonstrate that these four methods exhibit consistency in recovery from unlearning with those depicted in the main paper. Specifically, when confronted with instances initially misclassified by the MU models, after applying the CMIRA method, it is observed that the attack model can successfully restore the classification capability.

## D  ADDITIONAL ANALYSIS

Due to the length of the main paper, this part of the experimental results is not mentioned in the main body. The experimental results of this part aim to explore the factors that can affect CMIRA. We explore the influence of datasets, MU methods, and backbone networks on unlearning recovery attacks.

### D.1  IMPACT OF DIFFERENT DATASETS

We aggregated all CMIRA performance metrics within the same dataset to analyze how different datasets might influence their effectiveness. The results of this analysis are shown in Figure A3. The experimental results demonstrate that the diversity of datasets significantly influences the effectiveness of recovery attack. Specifically, TinyImageNet shows the highest recovery improvement and diversity, followed by Cifar-100, Cifar-10, and FMNIST. Higher dataset diversity, characterized by richer categories and greater sample differences, leads to more pronounced unlearning effects and a higher recovery rate. This suggests that while models trained on the more diverse dataset are more prone to forgetting, they are also at higher risk of induced recovery attacks.

### D.2  IMPACT OF DIFFERENT UNLEARNING METHODS

We aggregated CMIRA's recovery performance according to different unlearning methods, recording the recovery rate $\mathbf{R}_R$ and the area of membership recovery $\mathbf{A}_R$ for each. The results are displayed in Figure A4. The experimental results show that the recovery improvement $\mathbf{R}_I$ varies significantly with different MU methods. The highest to lowest recovery improvement ranking is IU, GA, FF,

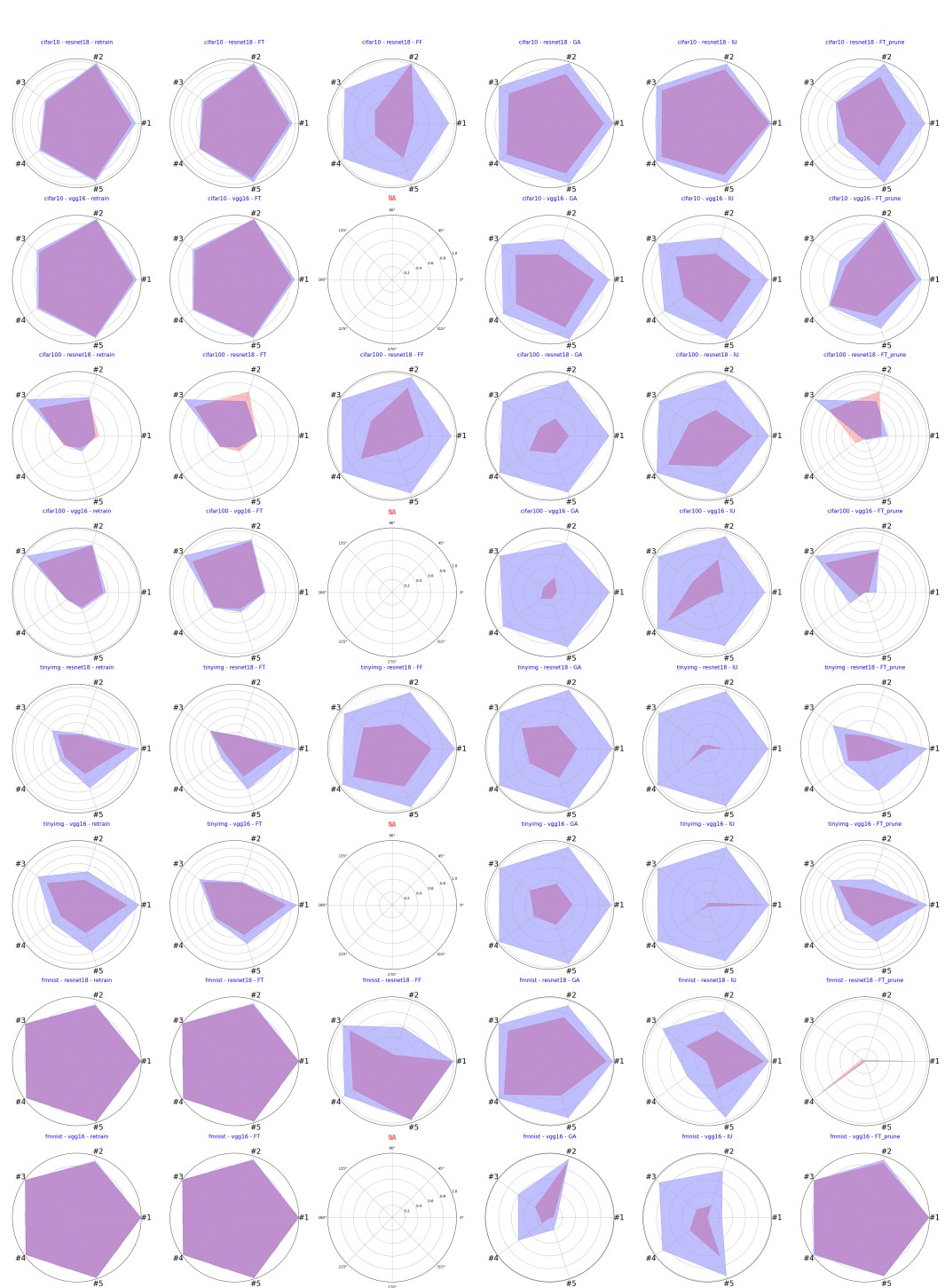

Figure A1: Membership Recovery Polygon (MRP). The red area represents the prediction accuracy of the unlearned model for each label, while the blue area represents the prediction accuracy after memory recovery by CMIRA. '#n' represents the n-th forgetted class.*The FT_Prune in the figure is referred to WP.*

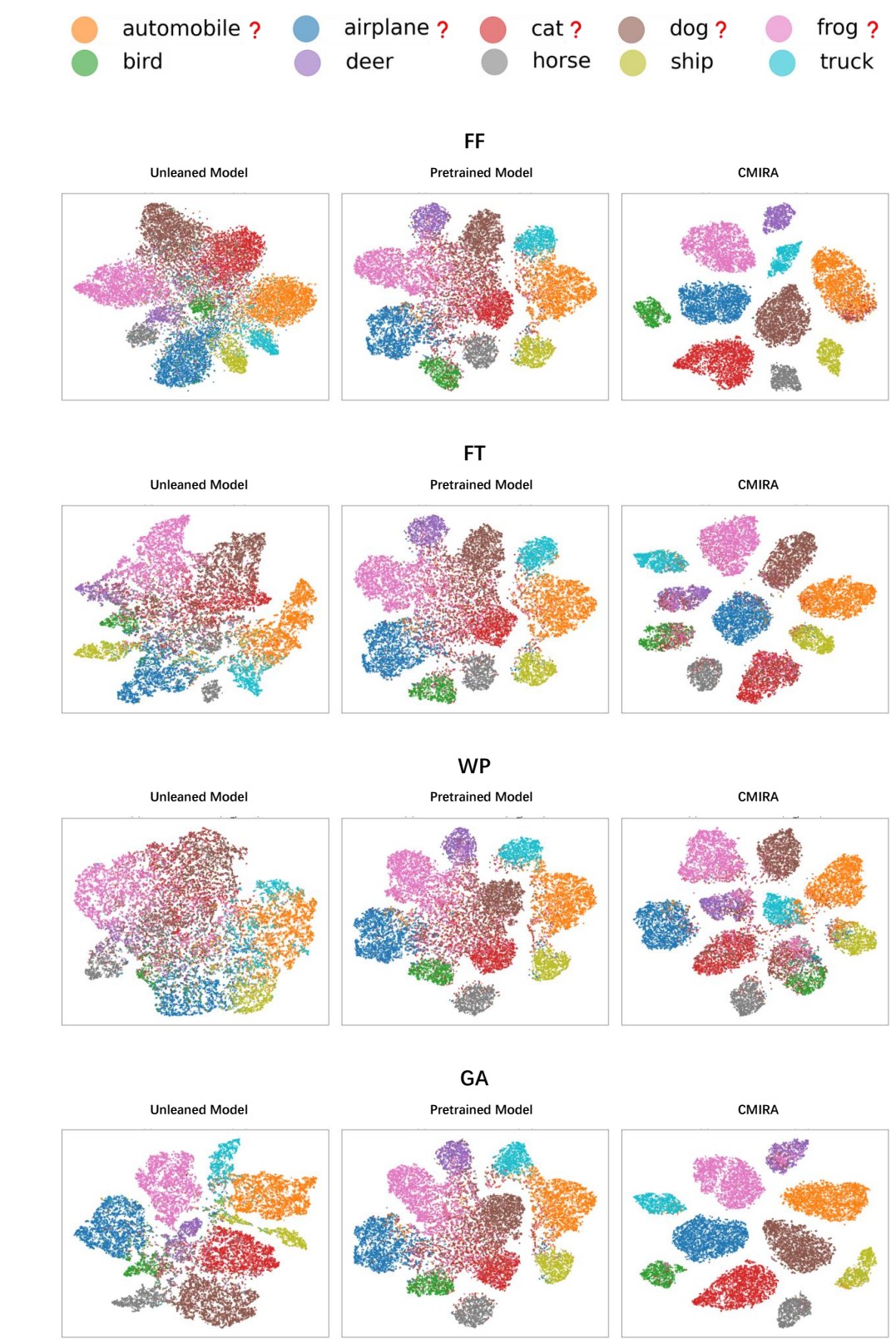

Figure A2: t-SNE plots of Cifar-10 datapoints in $\mathcal{D}_f$ and $\mathcal{D}_p$ w.r.t. unlearned models, pretrained attack models, and CMIRA models. The legend labels followed by a question mark indicate the forgetting classes. This figure shows the complete t-SNE figures of FF, FT, WP, and GA unlearning methods.

WP, FT, and RT. In particular, IU, GA, and FF are more susceptible to recovery attacks, indicating that their unlearning effects are relatively unstable. WP and FT perform moderately by adopting fine-tuning or pruning. RT is an exact unlearning method results in models with less forgotten privacy information, reducing the attack effects.

### D.3 Impact of Different Backbone Models

We aggregated CMIRA's recovery performance according to different backbone models, recording the recovery rate $\mathbf{R}_R$ and the area of membership recovery $\mathbf{A}_R$ for each. The results are displayed in Figure A5. The experimental results indicate that when the backbone is similar, the unlearning recovery effect is not significantly affected by the network architecture. However, it is observed that VGG16 is more susceptible than ResNet18. This can be attributed to the simpler convolutional structure of VGG16, which allows for more detailed feature adjustments, enhancing the unlearning recovery effect compared to the residual structure of ResNet18.

## E  Additional Ablation Study Results

This part provides a complete interpretation of the **Visualization of Confusion Matrix** section within the **Ablation Study** part in the main body of the article.

As shown in Table A4 and Table A5, as well as the confusion matrices in Figure A6 of the ablation studies, the complete CMIRA method consistently demonstrates the best performance in the vast majority of cases.

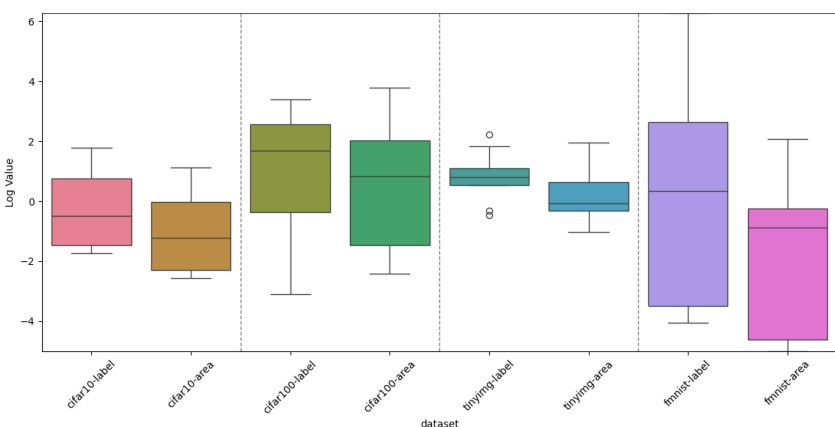

Figure A3: The boxplot compares the Recovery Rate and Area of Membership Recovery based on different datasets. Each section in the boxplot includes two indicators: the average value of the Recovery Rate ($\mathbf{R}_R$) grouped by dataset, and the average value of the Area of Membership Recovery ($\mathbf{A}_R$) grouped by dataset.

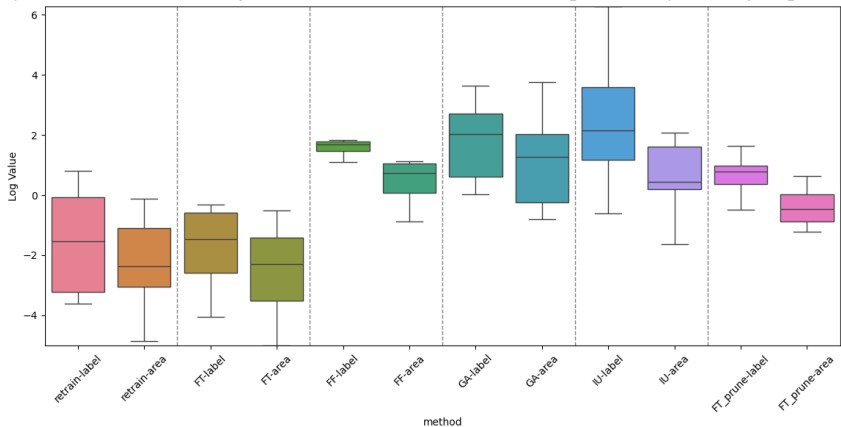

Figure A4: The boxplot compares the Recovery Rate and Area of Membership Recovery based on different machine unlearning (MU) methods. Each section in the boxplot includes two indicators: the average value of the Recovery Rate ($\mathbf{R}_R$) grouped by MU method, and the average value of the Area of Membership Recovery ($\mathbf{A}_R$) grouped by MU method. *The FT_Prune in the figure is referred to WP.*

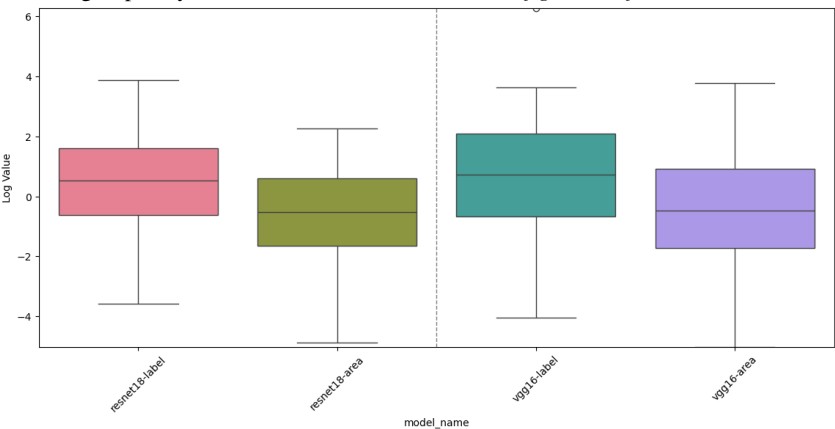

Figure A5: The boxplot compares the Recovery Rate and Area of Membership Recovery based on different backbone models. Each section in the boxplot includes two indicators: the average value of the Recovery Rate ($\mathbf{R}_R$) grouped by backbone model, and the average value of the Area of Membership Recovery ($\mathbf{A}_R$) grouped by backbone model.

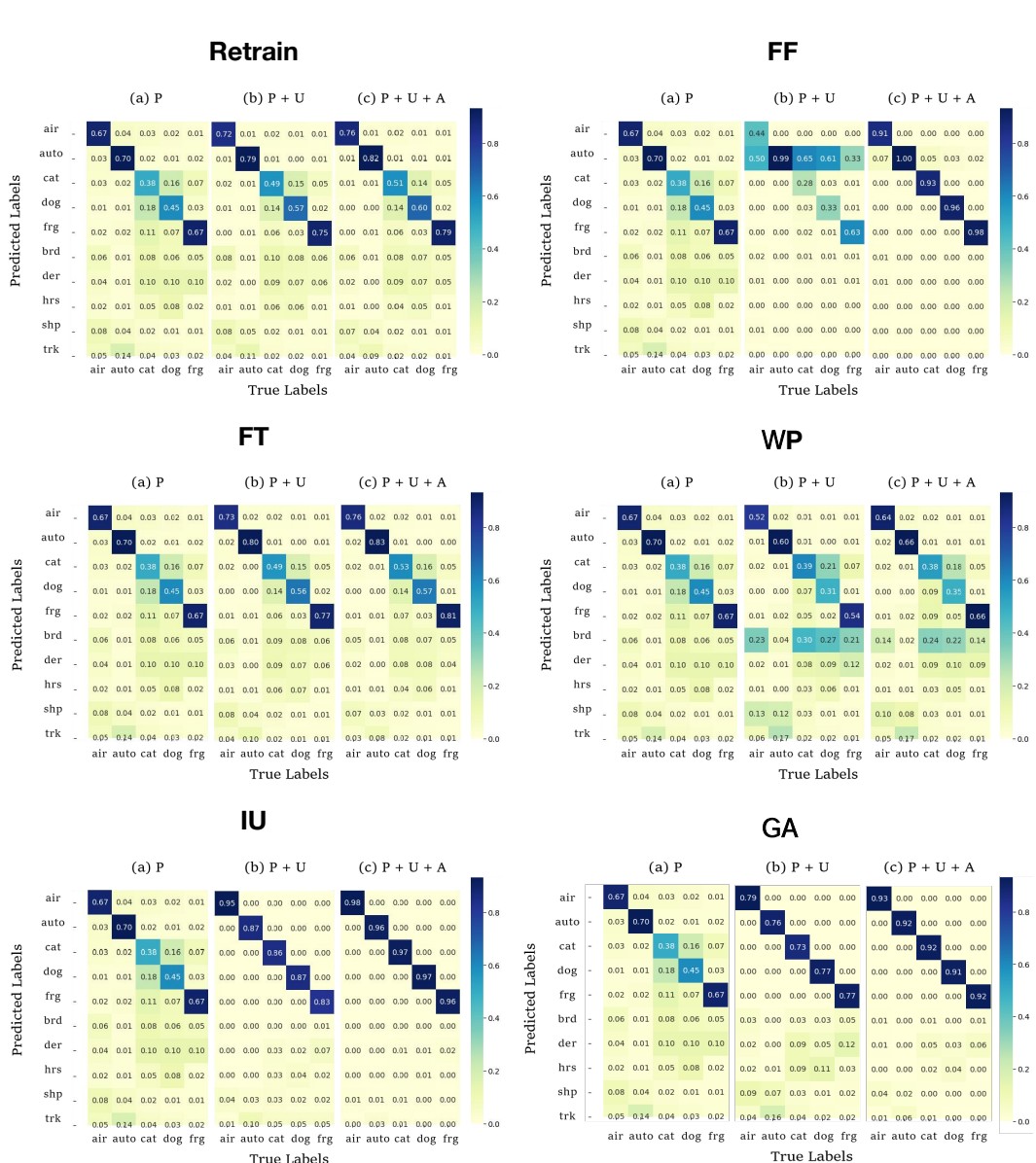

Figure A6: The plots of normalized confusion matrices demonstrate the classification performance of ablative models $\mathcal{M}^P$, $\mathcal{M}^{P+U}$, and $\mathcal{M}^{P+U+A}$ on Cifar-10 using the RT(Retrain)/FF/FT/WP/IU method respectively. The labels are reordered (the five forgetting classes are listed first) to better emphasize the class membership recovery capability achieved by CMIRA with the mutual knowledge distillation technique.

Table A3: Complete Class-wise evaluation on recovery attack efficacy of CMIRA (full performance). For the forgetting data of class #$i$, we present the $\mathcal{M}_U$'s prediction accuracy $\mathbf{Acc}_U^i$ (%) and the recovery amount achieved by CMIRA, displayed as $\uparrow(\mathbf{Acc}_A^i - \mathbf{Acc}_U^i)$.

| Dataset | Model | Method | Class #1
Airplane | Class #2
Automobile | Class #3
Cat | Class #4
Dog | Class #5
Frog | $\mathbf{A}_R$ (%) |
|---|---|---|---|---|---|---|---|---|
| Cifar-10 | ResNet18 | RT | 70.71 ↑ 5.51 | 79.07 ↑ 2.66 | 48.58 ↑ 2.35 | 57.11 ↑ 2.40 | 75.78 ↑ 2.84 | 11.20 |
| | | FT | 72.36 ↑ 4.13 | 80.58 ↑ 2.71 | 49.38 ↑ 3.55 | 55.96 ↑ 0.97 | 76.09 ↑ 5.11 | 10.54 |
| | | FF | 34.49 ↑ 56.58 | 99.91 ↑ 0.05 | 33.42 ↑ 60.00 | 33.16 ↑ 62.97 | 58.13 ↑ 39.60 | 314.98 |
| | | GA | 78.80 ↑ 14.44 | 75.64 ↑ 16.49 | 73.29 ↑ 18.58 | 77.16 ↑ 14.08 | 76.80 ↑ 15.38 | 44.68 |
| | | IU | 94.84 ↑ 3.60 | 86.76 ↑ 9.55 | 85.82 ↑ 11.02 | 86.62 ↑ 10.85 | 83.29 ↑ 13.02 | 20.31 |
| | | WP | 43.42 ↑ 20.45 | 51.69 ↑ 14.80 | 36.49 ↑ 1.95 | 24.84 ↑ 9.96 | 47.51 ↑ 18.93 | 84.69 |
| | VGG16 | RT | 81.07 ↑ 4.17 | 89.24 ↑ 1.52 | 65.96 ↑ 4.26 | 66.80 ↑ 3.51 | 85.24 ↑ 2.09 | 8.71 |
| | | FT | 81.91 ↑ 4.49 | 90.31 ↑ 0.98 | 69.56 ↑ 3.86 | 71.60 ↑ 2.27 | 85.64 ↑ 2.36 | 7.96 |
| | | GA | 64.36 ↑ 23.15 | 38.31 ↑ 23.29 | 61.07 ↑ 25.86 | 60.22 ↑ 24.00 | 72.93 ↑ 18.71 | 90.99 |
| | | IU | 62.71 ↑ 25.11 | 38.67 ↑ 24.97 | 55.64 ↑ 31.47 | 41.78 ↑ 35.11 | 64.84 ↑ 25.60 | 129.66 |
| | | WP | 72.71 ↑ 8.36 | 85.91 ↑ 4.22 | 33.60 ↑ 11.20 | 61.96 ↑ 2.13 | 55.33 ↑ 18.58 | 31.57 |

| Dataset | Model | Method | Class #1
Boy | Class #2
Clock | Class #3
Forest | Class #4
Lizard | Class #5
Otter | $\mathbf{A}_R$ (%) |
|---|---|---|---|---|---|---|---|---|
| Cifar-100 | ResNet18 | RT | 16.00 ↓ 2.67 | 27.56 ↑ 1.77 | 33.78 ↑ 11.11 | 11.56 ↓ 0.89 | 8.89 ↑ 2.67 | 16.74 |
| | | FT | 16.44 ↑ 0.45 | 34.67 ↓ 7.56 | 36.44 ↑ 9.78 | 13.33 ↓ 0.00 | 12.00 ↓ 3.11 | -2.24 |
| | | FF | 50.22 ↑ 44.00 | 79.56 ↑ 18.66 | 40.00 ↑ 58.67 | 60.89 ↑ 36.89 | 23.11 ↑ 72.45 | 279.63 |
| | | GA | 29.78 ↑ 64.00 | 28.44 ↑ 63.56 | 20.44 ↑ 71.12 | 38.67 ↑ 59.11 | 28.44 ↑ 64.89 | 942.53 |
| | | IU | 71.11 ↑ 26.67 | 42.67 ↑ 50.22 | 34.67 ↑ 59.55 | 76.89 ↑ 22.22 | 50.67 ↑ 46.22 | 182.81 |
| | | WP | 10.67 ↑ 4.00 | 30.22 ↓ 6.66 | 28.44 ↑ 11.56 | 7.56 ↓ 5.34 | 1.78 ↑ 0.89 | 6.07 |
| | VGG16 | RT | 24.89 ↑ 3.11 | 46.67 ↑ 0.89 | 46.22 ↑ 12.89 | 11.56 ↓ 0.00 | 14.67 ↑ 2.22 | 25.39 |
| | | FT | 28.89 ↑ 0.89 | 52.44 ↑ 1.78 | 50.67 ↑ 10.22 | 24.44 ↑ 0.89 | 16.89 ↑ 3.11 | 18.21 |
| | | GA | 10.67 ↑ 82.66 | 23.56 ↑ 56.88 | 11.56 ↑ 84.44 | 16.44 ↑ 73.34 | 9.78 ↑ 79.55 | 4378.30 |
| | | IU | 25.33 ↑ 67.11 | 55.11 ↑ 39.11 | 28.44 ↑ 69.34 | 78.22 ↑ 20.89 | 7.56 ↑ 82.66 | 708.15 |
| | | WP | 3.56 ↑ 7.55 | 42.22 ↑ 2.22 | 48.44 ↑ 12.00 | 6.67 ↑ 11.55 | 0.00 ↓ 0.00 | 74.04 |

| Dataset | Model | Method | Class #1
Salamander | Class #2
Baboon | Class #3
Hammer | Class #4
Umbrella | Class #5
Slug | $\mathbf{A}_R$ (%) |
|---|---|---|---|---|---|---|---|---|
| TinyImg | ResNet18 | RT | 56.00 ↑ 15.20 | 16.40 ↑ 1.20 | 26.80 ↑ 8.40 | 16.80 ↑ 6.40 | 30.00 ↑ 17.60 | 71.88 |
| | | FT | 57.20 ↑ 17.60 | 16.40 ↓ 0.40 | 36.40 ↓ 0.00 | 12.40 ↑ 6.40 | 35.20 ↑ 16.40 | 67.20 |
| | | FF | 62.40 ↑ 37.60 | 40.40 ↑ 54.00 | 56.00 ↑ 38.40 | 76.40 ↑ 21.20 | 63.20 ↑ 34.80 | 162.01 |
| | | GA | 43.20 ↑ 56.00 | 38.40 ↑ 59.20 | 54.40 ↑ 43.60 | 38.80 ↑ 60.00 | 48.00 ↑ 51.20 | 400.92 |
| | | IU | 23.60 ↑ 75.60 | 5.60 ↑ 92.40 | 9.60 ↑ 88.40 | 36.00 ↑ 64.00 | 0.00 ↑ 98.00 | 5273.35 |
| | | WP | 55.20 ↑ 32.80 | 18.00 ↑ 3.60 | 35.20 ↑ 20.80 | 28.80 ↑ 6.80 | 18.00 ↑ 44.80 | 186.25 |
| | VGG16 | RT | 60.00 ↑ 14.80 | 31.20 ↑ 10.80 | 44.00 ↑ 13.60 | 23.20 ↑ 13.20 | 35.20 ↑ 23.60 | 83.35 |
| | | FT | 63.20 ↑ 13.60 | 28.40 ↑ 0.80 | 47.60 ↑ 5.60 | 27.60 ↑ 2.80 | 38.40 ↑ 11.60 | 39.61 |
| | | GA | 36.40 ↑ 62.40 | 35.20 ↑ 62.40 | 39.20 ↑ 60.40 | 30.80 ↑ 69.20 | 32.80 ↑ 66.40 | 694.94 |
| | | IU | 93.20 ↑ 6.40 | 3.20 ↑ 95.20 | 0.00 ↑ 99.60 | 16.40 ↑ 82.80 | 2.40 ↑ 93.20 | 531.60 |
| | | WP | 70.80 ↑ 12.00 | 20.80 ↑ 14.80 | 43.20 ↑ 12.80 | 18.40 ↑ 14.40 | 29.20 ↑ 22.40 | 81.75 |

| Dataset | Model | Method | Class #1
Trouser | Class #2
Dress | Class #3
Sandal | Class #4
Sneaker | Class #5
Ankle boot | $\mathbf{A}_R$ (%) |
|---|---|---|---|---|---|---|---|---|
| FMNIST | ResNet18 | RT | 98.67 ↓ 0.19 | 90.00 ↑ 1.52 | 97.70 ↑ 0.26 | 95.56 ↑ 0.29 | 97.19 ↑ 0.11 | 0.64 |
| | | FT | 98.48 ↑ 0.04 | 91.11 ↑ 1.93 | 98.19 ↑ 0.33 | 96.52 ↑ 0.67 | 96.85 ↑ 0.04 | 1.06 |
| | | FF | 97.26 ↑ 2.07 | 10.52 ↑ 47.07 | 84.70 ↑ 13.67 | 77.70 ↑ 17.52 | 99.93 ↓ 1.04 | 40.69 |
| | | GA | 88.78 ↑ 10.52 | 72.41 ↑ 19.81 | 80.63 ↑ 18.00 | 88.48 ↑ 10.52 | 56.37 ↑ 37.22 | 51.41 |
| | | IU | 90.81 ↑ 8.08 | 50.70 ↑ 33.37 | 41.59 ↑ 47.22 | 0.85 ↑ 38.67 | 46.85 ↑ 48.11 | 118.82 |
| | | WP | 21.78 ↑ 54.07 | 0.00 ↑ 1.70 | 5.67 ↓ 4.86 | 94.48 ↑ 4.56 | 0.00 ↑ 0.37 | 497.19 |
| | VGG16 | RT | 98.78 ↓ 0.00 | 89.59 ↑ 2.11 | 98.11 ↓ 0.04 | 97.07 ↑ 0.04 | 97.19 ↑ 0.29 | 0.85 |
| | | FT | 98.67 ↑ 0.22 | 92.59 ↑ 1.37 | 98.85 ↑ 0.08 | 97.19 ↑ 0.33 | 97.41 ↑ 0.26 | 0.86 |
| | | GA | 5.52 ↑ 5.48 | 90.41 ↑ 5.55 | 27.78 ↑ 33.00 | 14.11 ↑ 46.56 | 0.67 ↑ 19.00 | 251.32 |
| | | IU | 0.04 ↑ 22.85 | 19.30 ↑ 57.92 | 20.85 ↑ 72.59 | 33.48 ↑ 53.63 | 63.48 ↑ 33.11 | 775.58 |
| | | WP | 96.41 ↑ 0.44 | 86.70 ↑ 4.97 | 94.89 ↑ 0.70 | 93.85 ↑ 2.30 | 92.63 ↑ 0.15 | 3.26 |

Table A4: Ablation Studies on ResNet18. All the metric scores are reported by (%). The accuracy of MU models $\text{Acc}_U$ in percentage (%) is reported as baseline. And P, P+U and P+U+A stand for the models $\mathcal{M}^P$, $\mathcal{M}^{P+U}$, and $\mathcal{M}^{P+U+A}$ respectively.

| Cifar-10 | RT | | | FT | | | FF | | | GA | | | IU | | | WP | | |
|---|---|---|---|---|---|---|---|---|---|---|---|---|---|---|---|---|---|---|
| | Acc | $R_R$ | $A_R$ | Acc | $R_R$ | $A_R$ | Acc | $R_R$ | $A_R$ | Acc | $R_R$ | $A_R$ | Acc | $R_R$ | $A_R$ | Acc | $R_R$ | $A_R$ |
| Baseline | 66.25 | - | - | 66.87 | - | - | 51.82 | - | - | 76.34 | - | - | 87.47 | - | - | 40.79 | - | - |
| P | 57.49 | -13.22 | -21.49 | 57.49 | -14.02 | -23.42 | 57.49 | 10.94 | 62.22 | 57.49 | -24.69 | -40.27 | 57.49 | -34.27 | -55.29 | 57.49 | 40.95 | 107.68 |
| P+U | 66.27 | 0.03 | 0.61 | 67.05 | 0.27 | 0.91 | 53.49 | 3.22 | 14.61 | 76.29 | -0.06 | -0.07 | 87.45 | -0.02 | -0.03 | 47.15 | 15.58 | 35.74 |
| **P+U+A** | 69.40 | 4.76 | 11.20 | 70.17 | 4.93 | 10.54 | 95.66 | 84.60 | 314.98 | 92.13 | 20.69 | 44.68 | 97.08 | 10.99 | 20.31 | 54.01 | 32.40 | 84.69 |

| Cifar-100 | RT | | | FT | | | FF | | | GA | | | IU | | | WP | | |
|---|---|---|---|---|---|---|---|---|---|---|---|---|---|---|---|---|---|---|
| | Acc | $R_R$ | $A_R$ | Acc | $R_R$ | $A_R$ | Acc | $R_R$ | $A_R$ | Acc | $R_R$ | $A_R$ | Acc | $R_R$ | $A_R$ | Acc | $R_R$ | $A_R$ |
| Baseline | 19.56 | - | - | 22.58 | - | - | 50.76 | - | - | 29.16 | - | - | 55.20 | - | - | 15.73 | - | - |
| P | 11.29 | -42.27 | -69.03 | 11.29 | -50.00 | -76.22 | 11.29 | -77.76 | -95.23 | 11.29 | -61.28 | -86.13 | 11.29 | -79.55 | -96.45 | 11.29 | -28.25 | -54.58 |
| P+U | 18.76 | -4.09 | -10.37 | 23.47 | 3.94 | 7.57 | 50.84 | 0.18 | 0.34 | 29.42 | 0.91 | 1.87 | 55.29 | 0.16 | 0.21 | 16.27 | 3.39 | 5.01 |
| **P+U+A** | 21.96 | 12.27 | 16.74 | 22.49 | -0.39 | -2.24 | 96.89 | 90.89 | 279.63 | 93.69 | 221.34 | 942.53 | 96.18 | 74.24 | 182.81 | 16.62 | 5.65 | 6.07 |

| TinyImg | RT | | | FT | | | FF | | | GA | | | IU | | | WP | | |
|---|---|---|---|---|---|---|---|---|---|---|---|---|---|---|---|---|---|---|
| | Acc | $R_R$ | $A_R$ | Acc | $R_R$ | $A_R$ | Acc | $R_R$ | $A_R$ | Acc | $R_R$ | $A_R$ | Acc | $R_R$ | $A_R$ | Acc | $R_R$ | $A_R$ |
| Baseline | 29.20 | - | - | 31.52 | - | - | 59.68 | - | - | 44.56 | - | - | 14.96 | - | - | 31.04 | - | - |
| P | 6.72 | -76.99 | -97.99 | 6.72 | -78.68 | -98.14 | 6.72 | -88.74 | -99.34 | 6.72 | -84.92 | -98.77 | 6.72 | -55.08 | -86.83 | 6.72 | -78.35 | -98.01 |
| P+U | 33.36 | 14.25 | 39.92 | 35.12 | 11.42 | 30.35 | 59.84 | 0.27 | 1.24 | 45.68 | 2.51 | 6.93 | 17.68 | 18.18 | 153.70 | 33.44 | 7.73 | 31.70 |
| **P+U+A** | 38.96 | 33.42 | 71.88 | 39.52 | 25.38 | 67.20 | 96.88 | 62.33 | 162.01 | 98.56 | 121.18 | 400.92 | 98.64 | 559.36 | 5273.4 | 52.80 | 70.10 | 186.25 |

| FMNIST | RT | | | FT | | | FF | | | GA | | | IU | | | WP | | |
|---|---|---|---|---|---|---|---|---|---|---|---|---|---|---|---|---|---|---|
| | Acc | $R_R$ | $A_R$ | Acc | $R_R$ | $A_R$ | Acc | $R_R$ | $A_R$ | Acc | $R_R$ | $A_R$ | Acc | $R_R$ | $A_R$ | Acc | $R_R$ | $A_R$ |
| Baseline | 95.82 | - | - | 96.23 | - | - | 74.02 | - | - | 77.33 | - | - | 46.16 | - | - | 24.39 | - | - |
| P | 94.19 | -1.71 | -3.19 | 94.19 | -2.12 | -3.81 | 94.19 | 27.24 | 51.95 | 94.19 | 21.79 | 44.46 | 94.19 | 104.03 | 179.13 | 94.19 | 286.24 | 5233.2 |
| P+U | 96.04 | 0.22 | 0.41 | 96.26 | 0.03 | 0.04 | 77.93 | 5.27 | 8.57 | 81.84 | 5.83 | 11.57 | 53.13 | 15.08 | 20.13 | 26.16 | 7.26 | 50.47 |
| **P+U+A** | 96.22 | 0.42 | 0.64 | 96.83 | 0.62 | 1.06 | 89.88 | 21.42 | 40.69 | 96.55 | 24.85 | 51.41 | 81.25 | 76.01 | 118.82 | 35.56 | 45.81 | 497.19 |

Table A5: Ablation Studies on VGG16. All the metric scores are reported by (%). The accuracy of MU models $\text{Acc}_U$ in percentage (%) is reported as baseline. And P, P+U and P+U+A stand for the models $\mathcal{M}^P$, $\mathcal{M}^{P+U}$, and $\mathcal{M}^{P+U+A}$ respectively.

| Cifar-10 | RT | | | FT | | | GA | | | IU | | | WP | | |
|---|---|---|---|---|---|---|---|---|---|---|---|---|---|---|---|
| | Acc | $R_R$ | $A_R$ | Acc | $R_R$ | $A_R$ | Acc | $R_R$ | $A_R$ | Acc | $R_R$ | $A_R$ | Acc | $R_R$ | $A_R$ |
| Baseline | 77.66 | - | - | 79.80 | - | - | 59.38 | - | - | 52.73 | - | - | 61.90 | - | - |
| P | 57.49 | -25.97 | -42.48 | 57.49 | -27.96 | -45.18 | 57.49 | -3.17 | -2.75 | 57.49 | 9.04 | 19.90 | 57.49 | -7.12 | -11.86 |
| P+U | 77.64 | -0.02 | -0.06 | 79.80 | -0.01 | 0.07 | 58.63 | -1.26 | -0.60 | 54.84 | 4.01 | 11.36 | 63.13 | 1.98 | 5.02 |
| **P+U+A** | 80.77 | 4.01 | 8.71 | 82.60 | 3.50 | 7.96 | 82.38 | 38.74 | 90.99 | 81.18 | 53.96 | 129.66 | 70.80 | 14.37 | 31.57 |

| Cifar-100 | RT | | | FT | | | GA | | | IU | | | WP | | |
|---|---|---|---|---|---|---|---|---|---|---|---|---|---|---|---|
| | Acc | $R_R$ | $A_R$ | Acc | $R_R$ | $A_R$ | Acc | $R_R$ | $A_R$ | Acc | $R_R$ | $A_R$ | Acc | $R_R$ | $A_R$ |
| Baseline | 28.80 | - | - | 34.67 | - | - | 14.40 | - | - | 38.93 | - | - | 20.18 | - | - |
| P | 11.29 | -60.80 | -86.00 | 11.29 | -67.44 | -90.19 | 11.29 | -21.60 | -35.84 | 11.29 | -71.00 | -89.40 | 11.29 | -44.05 | -72.31 |
| P+U | 28.00 | -2.78 | -5.56 | 34.84 | 0.51 | 0.45 | 13.87 | -3.70 | 9.86 | 38.49 | -1.14 | -1.17 | 19.56 | -3.08 | -13.07 |
| **P+U+A** | 32.62 | 13.27 | 25.39 | 38.04 | 9.74 | 18.21 | 89.78 | 523.46 | 4378.3 | 94.76 | 143.38 | 708.15 | 26.84 | 33.04 | 74.04 |

| TinyImg | RT | | | FT | | | GA | | | IU | | | WP | | |
|---|---|---|---|---|---|---|---|---|---|---|---|---|---|---|---|
| | Acc | $R_R$ | $A_R$ | Acc | $R_R$ | $A_R$ | Acc | $R_R$ | $A_R$ | Acc | $R_R$ | $A_R$ | Acc | $R_R$ | $A_R$ |
| Baseline | 38.72 | - | - | 41.04 | - | - | 34.88 | - | - | 23.04 | - | - | 36.48 | - | - |
| P | 6.72 | -82.64 | -98.67 | 6.72 | -83.63 | -98.80 | 6.72 | -80.73 | -98.06 | 6.72 | -70.83 | -98.45 | 6.72 | -81.58 | -98.67 |
| P+U | 38.56 | -0.41 | 3.12 | 40.80 | -0.58 | 0.04 | 38.24 | 9.63 | 31.40 | 22.56 | -2.08 | -3.63 | 36.56 | 0.22 | -0.30 |
| **P+U+A** | 53.92 | 39.26 | 83.35 | 47.92 | 16.76 | 39.61 | 99.04 | 183.94 | 694.94 | 98.48 | 327.43 | 531.60 | 51.76 | 41.89 | 81.75 |

| FMNIST | RT | | | FT | | | GA | | | IU | | | WP | | |
|---|---|---|---|---|---|---|---|---|---|---|---|---|---|---|---|
| | Acc | $R_R$ | $A_R$ | Acc | $R_R$ | $A_R$ | Acc | $R_R$ | $A_R$ | Acc | $R_R$ | $A_R$ | Acc | $R_R$ | $A_R$ |
| Baseline | 96.15 | - | - | 96.94 | - | - | 27.70 | - | - | 27.43 | - | - | 92.90 | - | - |
| P | 94.19 | -2.04 | -3.77 | 94.19 | -2.84 | -5.04 | 94.19 | 240.06 | 1462.7 | 94.19 | 243.37 | 1567.6 | 94.19 | 1.39 | 2.74 |
| P+U | 96.20 | 0.05 | 0.08 | 97.00 | 0.06 | 0.08 | 29.98 | 8.24 | 20.40 | 35.71 | 30.19 | 74.54 | 93.87 | 1.04 | 1.90 |
| **P+U+A** | 96.63 | 0.50 | 0.85 | 97.39 | 0.47 | 0.86 | 49.61 | 79.14 | 251.32 | 75.45 | 175.07 | 775.58 | 94.61 | 1.84 | 3.26 |

