# OpenReview forum: "CMIRA: Class Membership Inducing Recovery Attacks Against Machine Unlearning Models"
_ICLR.cc/2025/Conference — ICLR 2025 Conference Withdrawn Submission_

### Official Review · Reviewer_ueKf · 2024-10-23

**Soundness:** 2
**Presentation:** 3
**Contribution:** 2
**Rating:** 3
**Confidence:** 4

**Summary:**

The implementation of data privacy regulations such as GDPR and CCPA has propelled advancements in machine unlearning (MU) technology, which enables the removal of specific sensitive data points from trained models upon request. This paper reformulates the study of attacks against released unlearned models and presents the first exploration of recovery attacks on MU models without requiring access to the original model. The proposed approach, called Class Membership Inducing Recovery Attack (CMIRA), effectively recovers forgotten data by leveraging a probing dataset.

**Strengths:**

1.	The paper introduces CMIRA, a privacy attack scheme, which can effectively recover the true class memberships from commonly used MU models.
2.	The authors successfully implement a recovery attack model using mutual knowledge distillation, leveraging a probing dataset to recover class memberships from MU models.
3.	The paper conducts extensive experiments across four widely used datasets in MU research

**Weaknesses:**

1.	The authors claim that " It is imperative that the originally learned model is inaccessible to users," but this viewpoint may be questionable. Similar to a scenario where a user logs into a website, the server could train the model using the user's data during the session and perform data unlearning after the user logs out. However, before the unlearning occurs, other users can still access the model provided by the website.
2.	The literature review is insufficient. The authors state that " The only existing research (Hu et al., 2024) that investigates attacks against MU models was recently published," but in fact, several studies in recent years have explored the privacy issues of unlearned data, as outlined below:
- [CCS’21] When machine unlearning jeopardizes privacy.
- [Arxiv 2024] Inexact unlearning needs more careful evaluations to avoid a false sense of privacy.
- [IJIS’22] Label-only membership inference attacks on machine unlearning without dependence of posteriors.
3.	The authors mention that the distribution of the Probing Dataset should match that of the forgotten dataset—does this also include the label distribution? If so, an attacker could infer the labels of the forgotten data directly from the distribution.
4.	Algorithm 1 assumes that the attacker knows the parameters of the unlearned model, which seems unrealistic. In real-world scenarios, black-box environments are more common.
5.	In the experimental section, if only 5% of the data is forgotten, would the attack still be effective?

**Questions:**

Check the weakness above.

**Details Of Ethics Concerns:**

None.

---

### Official Review · Reviewer_broq · 2024-10-29

**Soundness:** 2
**Presentation:** 3
**Contribution:** 2
**Rating:** 5
**Confidence:** 4

**Summary:**

This paper proposes an attack method in the field of machine unlearning, termed Class Membership Inducing Recovery Attack (CMIRA). The authors define CMIRA as an attack technique that recovers the utility of forgetting data from an unlearned model. A notable strength of this approach is that, unlike previous works, it does not require knowledge of the unlearned model's parameters, making it a more practical black-box method. Experimental results demonstrate the effectiveness of this method by successfully attacking models produced by several unlearning techniques.

**Strengths:**

- This paper addresses an interesting and timely topic. With the enactment of various laws and regulations on the protection of personal privacy, the issue of machine unlearning has garnered significant attention from industry and academia. However, defenses for unlearned models have largely been overlooked. By approaching this from an attack perspective, the authors demonstrate that current unlearning techniques leave unlearned models partially vulnerable to attacks.
- Building on previous work [1], this study proposes an attack method that does not require access to the parameters of the unlearned model, making CMIRA a more practical and applicable black-box attack. This amplifies the risk of attacks on unlearned models.
- The experimental section of this study is comprehensive, validating the effectiveness of the method from multiple experimental perspectives. The inclusion of experimental code also enhances the reproducibility and transparency of the approach.

[1] Hongsheng Hu, Shuo Wang, Tian Dong, and Minhui Xue. Learn what you want to unlearn: Unlearning inversion attacks against machine unlearning. In IEEE Symposium on Security and Privacy, 2024.

**Weaknesses:**

- The initial assumptions underlying the proposed solution do not entirely align with practical scenarios. The paper defines the attack as compromising the privacy of forgotten data; however, the definition of privacy varies across different contexts and is not limited to correct predictions. For instance, in class forgetting (such as in a facial recognition system), forgetting a user’s face means that the unlearned model can no longer predict that user’s information. In this context, can the dataset Dp used in the paper truly be obtained easily? If so, then privacy would already be compromised during the data collection phase, as facial data are inherently private.
- Some expressions and definitions in the paper lack clarity, such as "class membership" and the phrase "inspired by MIA"  in page 1, line 52. Does this refer to gathering auxiliary datasets with distributions similar to MIA methods?
- The comparison algorithms chosen are somewhat outdated. Methods such as RT, FT, and GA are only basic, relatively unsuccessful unlearning techniques that have not achieved strong unlearning benchmarks, especially in terms of privacy (e.g., ASR in MIA). Among recent methods, only WP has been included from the past three years.

**Questions:**

Q1. Is the dataset Dp composed of an entire class of data, or is it a subset randomly drawn from all classes? Based on the composition of the forgetting data, unlearning methods can be categorized into class unlearning and sample unlearning. It is not explicitly clear in the text which category applies to this study, or whether it aims to address both. Additionally, have the differing collection requirements for Dp in each unlearning scenario been considered?

Q2. The attack on the unlearned model presented here resembles a verification method for assessing the effectiveness of unlearning, much like membership inference attacks (MIA) serve as a privacy metric. High attack success rates could indicate a lack of unlearning efficacy or signal unsuccessful unlearning. Is there a fundamental difference between this viewpoint and the attack approach detailed in this paper?

Q3. On page 5, line 263, the paper mentions that "the problem can be naturally formulated within the framework of semi-supervised learning (SSL)." For the inducing recovery stage, could alternative SSL solutions be applied, or does Mixup offer specific advantages in this context?

---

### Official Review · Reviewer_y7tC · 2024-10-31

**Soundness:** 2
**Presentation:** 3
**Contribution:** 1
**Rating:** 3
**Confidence:** 4

**Summary:**

This paper explores the vulnerabilities of machine unlearning (MU) technology, which is developed to comply with privacy regulations by removing specific sensitive data from trained models. The authors introduce a novel attack method, the Class Membership Inducing Recovery Attack (CMIRA), which recovers unlearned data without needing access to the original model. CMIRA leverages mutual knowledge distillation between MU and attack models, demonstrating effectiveness across various datasets and MU methods.

**Strengths:**

1. The diagrams are clear and effectively illustrate the methods used, making it easy to understand the approach.

2. The experiments are very comprehensive. The experimental figures are visually appealing.

3. The writing is easy to follow.

**Weaknesses:**

1. The definition of membership seems inaccurate. Typically, membership refers to whether a sample belongs to a dataset, such as the training or forget dataset. CMIRA, however, targets the labels of these data points after their membership in the forget dataset has been confirmed.
2. The description of unlearning, particularly “M_U has forgotten the true labels Y_f of X_f,” is misleading. Machine unlearning usually requires that the model was never trained on X_f.
3. The problem formulation is confusing. If the goal is to recover labels for X_f, wouldn’t simply labeling the data artificially solve this? Since you have already labeled a probing dataset, it’s unclear why the unlearned model is needed for this task or why this approach constitutes a privacy risk.

4. The experiments should show the model’s performance both before and after unlearning, including metrics commonly used in MU evaluations, like test accuracy, training accuracy, unlearning accuracy, and membership inference efficacy.

5. The notational complexity is hard to follow. Symbols such as P in line 245 and subsequent \textbf{p}, as well as numerous \hat{}, \bar{}, and \tilde{} notations, are difficult to understand without clearer explanations.

**Questions:**

1. Could you briefly explain how mutual knowledge distillation enables the unlearned model to recover predictions on the unlearned dataset? It appears that the attack model itself has some predictive ability on the unlearned dataset, acting almost like an Oracle by providing labels for the unlearned model to learn from.

2. What exactly do you mean by the "vulnerability of current MU-based privacy preservation"? If it refers to the model’s ability to learn labels of forgotten data, isn't that just a function of its learning capability, rather than a vulnerability?

---

### Official Review · Reviewer_fgM5 · 2024-11-01

**Soundness:** 2
**Presentation:** 2
**Contribution:** 2
**Rating:** 3
**Confidence:** 4

**Summary:**

This paper proposes an attack designed to infer the labels of unlearned samples in machine unlearning settings. The authors introduce a method that leverages a probing dataset, which is similar to the unlearned data, to predict the labels of unlearned samples. Experimental results demonstrate that CMIRA can recover labels of the unlearned data.

**Strengths:**

- Exploring vulnerability in machine unlearning settings is an interesting topic.

**Weaknesses:**

- The presentation could be improved for clarity.
- The motivation behind the problem being studied could be clearer.
- The assumptions underlying the attack require more detailed explanation.

**Questions:**

I suggest the authors adopt commonly accepted notation for better clarity. Currently, the symbols used differ from standard conventions, which may cause confusion. For instance, datasets should not be denoted as mappings between data and outputs, as they are collections rather than functions.

The motivation behind this problem could be clearer. The goal is to infer labels of unlearned samples, but does this have practical implications or pose actual privacy risks? If the attacker has access to the unlearned samples, they could likely label these samples independently, so it is unclear why an attack is necessary in this scenario.

The assumptions for the attack scenario also need more detail. The paper seems to assume that the unlearned model cannot correctly classify the unlearned samples, as they are no longer part of the training set. However, this assumption may not hold unless the model is highly overfitted and lacks generalization to any unseen data.

Based on the attack description, it seems that the attacker does not directly extract information from the target model; instead, the label information comes from the probing dataset. This reinforces the question about the practical privacy implications of this attack: if the knowledge is derived independently of the target model, does this really constitute a privacy threat?

---

### Official Review · Reviewer_ksRU · 2024-11-04

**Soundness:** 2
**Presentation:** 4
**Contribution:** 3
**Rating:** 5
**Confidence:** 5

**Summary:**

This paper presents a class membership inference attack targeting machine unlearning pipelines. Unlike prior work, the authors assume that only the unlearned model is accessible to the attacker. The attack’s efficacy is evaluated across four classical datasets and two deep neural networks, demonstrating success on six unlearning pipelines. This attack poses a privacy threat, and it also serves as a valuable tool for evaluating privacy leakage in practical applications.

**Strengths:**

- The paper is well-written and easy to follow, with sufficient background information and a clear explanation of the attack methodology.
- The attack approach is novel, as this work is the first to consider the scenario where the attacker has access only to the unlearned model.

**Weaknesses:**

- The attacker is assumed to know the unlearning data distribution, which may be unrealistic in practical settings. The authors should justify this assumption and discuss the potential impact on attack performance if the attacker lacks this knowledge.
- Although a successful attack highlights a real privacy threat, the authors should expand on this risk by discussing potential mitigation methods to reduce potential misuse of the proposed attack.
- Additionally, in the context of privacy attacks against machine unlearning, the authors should consider adding the following relevant references:

[a] When Machine Unlearning Jeopardizes Privacy. ACM CCS 2022.

[b] How to Combine Membership-Inference Attacks on Multiple Updated Machine Learning Models. PoPETs 2023.

**Questions:**

1. Why is it more practical to assume the attacker can access only the unlearned model rather than both versions of the model?
2. Could you elaborate on the rationale behind assuming the attacker has knowledge of the unlearning data distribution?
3. Are there any potential mitigation methods for this privacy risk?
4. In Table 1, why is the result of the FF unlearning method on the VGG16 model missing?

**Details Of Ethics Concerns:**

This paper proposes a privacy attack against machine unlearning pipelines, highlighting a potential threat to machine learning models. To prevent potential misuse, the authors should discuss possible mitigation strategies for this privacy risk.

---

### Note · Authors · 2024-11-20

I have read and agree with the venue's withdrawal policy on behalf of myself and my co-authors.